# Trajectories of self-kindness, common humanity, and mindfulness during the COVID-19 pandemic: A person-oriented multi-trajectory approach

Robin Wollast[1]*, Éric Lacourse[2], Geneviève A. Mageau[1], Mathieu Pelletier-Dumas[1], Anna Dorfman[3], Véronique Dupéré[4], Jean-Marc Lina[5], Dietlind Stolle[6], Roxane de la Sablonnière[1]

1 Department of Psychology, University of Montreal, Montreal, Quebec, Canada, 2 Department of Sociology, University of Montreal, Montreal, Quebec, Canada, 3 Department of Psychology, Bar-Ilan University, Ramat Gan, Israel, 4 School of Psychoeducation, University of Montreal, Montreal, Quebec, Canada, 5 École de Technologie Supérieure, Université du Québec, Montreal, Quebec, Canada, 6 Department of Political Science and Centre for the Study of Democratic Citizenship, McGill University, Montreal, Quebec, Canada

* robin.wollast@hotmail.com, roxane.de.la.sablonniere@umontreal.ca

## Abstract

The COVID-19 pandemic has produced unprecedented changes in the lives of many people. Although research has documented associations between concerns related to COVID-19 and poor mental health indicators, fewer studies have focused on positive factors that could help people better cope with this stressful social context. To fill this gap, the present research investigated the trajectories of self-compassion facets in times of dramatic social change. Using a longitudinal research design, we described the trajectories of self-kindness, common humanity, and mindfulness during the first eight months of the COVID-19 pandemic, in a representative sample of Canadian adults ($N = 3617$). Relying on a multi-trajectory group-based approach, we identified clusters of individuals following persistently low (4.0%), moderate-low (39.3%), moderate-high (46.7%), and high (10.0%) levels of self-kindness, common humanity, and mindfulness. Interestingly, we found that compassionate self-responding trajectories were mainly stable over time with minor fluctuations for some groups of individuals, in line with the epidemiological situation. In terms of covariates, we observed that older women were more likely to follow trajectories of high compassionate self-responding, as compared to the other age and gender groups. In terms of mental health indicators, we demonstrated that trajectory groups with high levels of compassionate self-responding were associated with greater life satisfaction, more happiness, better sleep quality, higher sleep quantity, and fewer negative emotions, as compared to lower trajectory groups. The results supported the idea that self-compassion during the COVID-19 pandemic could have favored better mental health indicators and could possibly be promoted as a psychological intervention in the general population.

**Data Availability Statement:** Questionnaire, raw data, code, and additional elements are publicly available on OSF at https://osf.io/t3xzy/.

**Funding:** This study is part of a larger national research project financed by the Canadian Institutes of Health Research (CIHR) that examines the social consequences of the COVID-19 pandemic on Canadians (grant number = 170633, for more information, see https://csdc-cecd.wixsite.com/covid19csi?lang=en). This project is also financed by the Social Sciences and Humanities Research Council (SSHRC). The funders had no role in study design, data collection and analysis, decision to publish, or preparation of the manuscript.

**Competing interests:** The authors have declared that no competing interests exist.

# Introduction

The COVID-19 pandemic has led to a dramatic social change that has instigated unprecedented transformations in the lives of billions of people. During this challenging time, the increasing number of COVID-19 cases, risk of infection, economic impact, social isolation, overburdening of health care, fear for loved ones, and other concerns about the pandemic contributed to feelings of psychological distress for many people worldwide (e.g., [1]). Although many studies have investigated the impact of the COVID-19 pandemic on mental health indicators (e.g., [2]), the key factors that might have helped people better cope with the challenges associated with the COVID-19 crisis are largely unknown. To address this gap, the present paper focuses on compassionate self-responding during the COVID-19 pandemic, as the ability to show compassion to oneself while suffering as a putative positive coping strategy of key mental health indicators.

The purpose of the present research was threefold: (1) identifying trajectories of compassionate self-responding (i.e., self-kindness, common humanity, and mindfulness) during the first eight months of the COVID-19 pandemic based on 10 waves of data collection (April—November 2020), (2) testing whether these trajectories are associated with sociodemographic and COVID-19-related variables, and (3) comparing these compassionate self-responding trajectories on key mental health indicators assessed 10 weeks after the main survey, in a large representative sample of Canadians.

## Mental health indicators and COVID-19

The COVID-19 pandemic and its consequences have negatively affected mental health worldwide. In a cross-national study involving 48 countries, scholars found high levels of negative emotions in the general population, with higher levels among women, young adults and those who expressed concern about getting infected [3]. Consistent with these findings, several other studies have documented the negative impact of COVID-19 on emotional experiences. Based on the pooled effect sizes of 19 studies, Ernst et al. [2] found an overall small increase in loneliness since the start of the pandemic (see also [4] for similar results). Similarly, in their systematic review and meta-analysis, Salari et al. [1] highlighted the prevalence of stress, anxiety, and depression among the general population during the COVID-19 pandemic. In the same vein, two meta-analyses demonstrated that the COVID-19 pandemic increased negative emotions and disrupted sleep quality [5, 6]. Moreover, Satici et al. [7] found that the fear of COVID-19 was positively associated with psychological distress and negatively associated with life satisfaction. Other studies have demonstrated an increase in psychological and mental health disorders globally during the COVID-19 pandemic [8].

The negative impact of COVID-19 on mental health indicators called for studies on psychological coping strategies that could have buffered this detrimental impact of this dramatic social change [9]. Understanding the mechanisms that could help improve different facets of mental health among individuals is also critical for developing preventive interventions and public policies to deal with future social crises. Accordingly, the present study investigated the temporal evolution (i.e., stability or change) in compassionate self-responding during the COVID-19 pandemic, as a key cognitive and personal dimension that could have buffered the pandemic's negative impact, providing a nuanced view of the link between this potential protective factor and mental health indicators. Based on the reviewed studies, we focused on participants' emotional well-being, happiness, and life satisfaction, which are commonly used to assess subjective well-being [10], as well as sleep quantity and quality, as indirect measures of psychological distress [11].

## Self-compassion

A growing body of research (e.g., [12–17]) suggested that responding in a self-compassionate way when struggling with negative events related to the COVID-19 pandemic, could have alleviated the stress and potential negative outcomes that these events could trigger (see also [18–20]). Responding in a self-compassionate way—or compassionate self-responding—refers to a person's ability to kindly accept oneself or show self-directed kindness while suffering [21]. It can thus be defined as a warm and non-judgmental attitude toward oneself during setbacks, which in turn could modulate one's adjustment to pandemic-related threats. Compassionate self-responding comprises three interconnected facets: self-kindness, the perception of personal experience as common to humanity, and mindfulness, which together with their opposite negative facets (i.e., self-judgment, isolation, and over-identification) form the global construct of self-compassion [22].

According to Neff [22], self-kindness entails being warm and understanding toward ourselves when we suffer, rather than ignoring our pain or flagellating ourselves with self-criticism. The pandemic changed the ways in which individuals relate to and navigate the world. Adopting a benevolent and kind attitude toward our reactions and behaviors toward and during this crisis was a personal attitude that may have protected our psychological resources [23]. Specifically, when our reality is accepted with benevolence and kindness, greater emotional equanimity is expected [23].

In the same vein, common humanity, the second positive facet of self-compassion, has been found to help people cope with adverse experiences [22]. Experiencing losses or failings can often be associated with a sense of isolation, as people come to believe that these sufferings are unique to them. When people recognize that life challenges are part of a shared human experience, they feel a sense of connectedness that can sooth suffering. Thus, recognizing that failings and sufferings experienced during the COVID-19 pandemic were part of a shared human experience, something that most people went through, rather than only their own [22], may have enabled another tool that could helped cope better with COVID-19-related challenges.

Finally, the third positive facet of self-compassion is mindfulness—the willingness to observe difficult emotions, thoughts, and behaviors with openness and clarity leading to a more balanced perspective regarding these emotions [24]. In the context of the COVID-19 pandemic, mindfulness entailed a non-judgmental, receptive mind-state in which one acknowledged thoughts and feelings as they were, without trying to suppress or deny them and without being trapped in emotional turmoil. Adopting a mindful perspective toward COVID-19-related events may have led, for example, to a more balanced approach to the experience of negative emotions in this context.

Prior research has demonstrated that self-compassion, comprised of both its positive and negative facets, is associated with less psychopathology (see [25] for a meta-analysis) and with higher emotional well-being (see [26] for a meta-analysis). A few cross-sectional studies have also demonstrated that self-compassion may have helped reduce people's intolerance for uncertainty and fear of COVID-19 [13], buffered adverse mental health impacts of COVID-19-related threats [20], and reduced symptoms of depression, anxiety, and stress during quarantine [12]. Regarding our mental health indicators of interest, self-compassion has been found to be cross-sectionally associated with less negative emotions [27, 28], more happiness [29], increased life satisfaction [30], and better sleep quality [11]. These findings suggested that adopting a more self-compassionate attitude when responding to the concerns and challenges related to the COVID-19 pandemic could have helped individuals cope with these challenges, resulting in positive mental health indicators. However, these studies examined the combined effects of compassionate self-responding and their negative counterparts rather than focusing

solely on compassionate self-responding. It is thus not clear if these associations would still be observed uniquely for self-compassion's positive facets ([31] but see also [32]).

In the current study, we focused on positive aspects of self-compassion during the COVID-19 pandemic because positive aspects that foster kindness and acceptance toward the self tend to play a major role in self-compassion interventions [33]. Also, drawing on past research on experience-based variability in other cognitions such as self-concept, attitudes, or global motivation ([34–37]), we reasoned that people are likely to vary in their compassionate self-responding across different contexts, depending on their past experiences [38]. Accordingly, we focused on compassionate self-responding while facing COVID-19 events specifically to obtain more precise measures of people's reactions to the COVID-19 pandemic.

Most studies on self-compassion during the COVID-19 pandemic have relied on cross-sectional methods (e.g., [12] but see also [17]). Although such approaches provide valuable information on between-groups differences in a single time-point, they may obscure critical variation in compassionate self-responding across individuals over time. In line with Neff et al.'s [39] theory, we expected strong and positive associations between trajectories of self-kindness, common humanity, and mindfulness, suggesting that these three facets of self-compassion would follow a similar pattern over time. Yet, we also expected that specific groups of individuals would demonstrate different levels of compassionate self-responding (i.e., low, moderate, and high) across the three indicators and over time. Specifically, we expected that low compassionate self-responding trajectory groups would show a non-linear shape, whereas high compassionate self-responding trajectory groups would be more stable over time, suggesting that when compassionate self-responding is not fully developed, it is likely to be more dependent on context.

To test these hypotheses, we used a longitudinal research design to investigate the trajectories of compassionate self-responding to COVID-19-related events during the first eight months of the pandemic using a person-oriented analytical method. Specifically, we conducted a multi-trajectory latent class growth analysis (LCGA) to identify homogeneous subgroups within the larger heterogeneous population that followed specific trajectories that could vary in prevalence and shape across the three positive self-compassion indicators [40, 41].

Scholars have evidenced that self-compassion was strongly associated with mental health among adolescents as well as adults [42]. Given that young adults tended to report significantly poorer mental health during the COVID-19 pandemic as compared to all other age groups (e.g., [43, 44]), one may argue that young adults also experienced lower levels of self-compassion during the pandemic and suffered more from its negative consequences. This is consistent with past research showing that older people are less likely to deny their problems and more likely to seek treatment for mental health difficulties [45]. Based on these findings, we formulated a tentative hypothesis that older adults would report greater levels of compassionate self-responding, compared to their younger counterparts.

In addition, it has been found that the COVID-19 pandemic has affected women differently than it has affected most men (e.g., [46, 47]). Specifically, women have reported more sleep problems and more symptoms of anxiety and depression (e.g., [48, 49]). These gender differences in well-being may have manifested because women were over-represented in essential frontline services, faced a higher risk of unemployment, experienced a greater burden of caring for childcare and other family members, and were more prone to experience gender-based violence during lockdown [50]. Taken together with research showing that men report slightly higher self-compassion than women ($d$ = .18; [51, 52]), it may suggest that women would have been more likely to follow trajectories characterized by low compassionate self-responding during the COVID-19 context, as compared to men.

## Aims and hypotheses

The present study analyzed heterogeneity in compassionate self-responding during a period of eight months following the beginning of the COVID-19 pandemic among a large representative sample of Canadians ($N$ = 3617 at Wave 1) and aimed at establishing preliminary evidence for causal inferences [41] on mental health indicators assessed 10 weeks later. The first goal of this longitudinal research was to identify groups of individuals who followed specific temporal dynamics in terms of levels (i.e., low, moderate, high) and shape (i.e., linear, quadratic, or cubic) of three compassionate self-responding indicators over time (assessed with single indicators). The second goal was to test whether these identified trajectory groups were associated with different sociodemographic and COVID-19 related variables. Finally, the third goal was to analyze whether compassionate self-responding trajectories were associated with several mental health indicators.

**Hypothesis 1.** Was compassionate self-responding heterogeneous or homogeneous in a representative sample of Canadians? We hypothesized that several groups with specific trajectories would be identified, differing in terms of their magnitude (i.e., low, moderate, and high levels of self-kindness, common humanity, and mindfulness).

**Hypothesis 2.** Were sociodemographic characteristics and COVID-19-related variables associated with different compassionate self-responding trajectories? We expected that multi-trajectory groups would be associated with age and gender. Specifically, we hypothesized that women and young adults would be more likely to follow trajectories characterized by low compassionate self-responding. Similarly, we expected that people who were most negatively affected by the pandemic (e.g., those more afraid of being infected) would be more likely to respond with self-compassion and thus belong to high compassionate self-responding trajectories.

**Hypothesis 3.** Did people who present different compassionate self-responding trajectories differ in various mental health indicators? We hypothesized that high multi-trajectory groups would be positively associated with indicators of good mental health (emotional well-being, happiness, sleep quality and quantity, and life satisfaction) assessed 10 weeks after the main survey.

## Method

This study was part of a larger longitudinal project, "COVID-19 Canada: The end of the world as we know it?" (see [53, 54]). The present manuscript pioneers the exploration of the positive facets of self-compassion, specifically compassionate self-responding trajectories, utilizing a three-step approach. Building on the work described in the current manuscript, we extended and successfully replicated the findings by relying on a global score of self-compassion (see [54]). This study complies with the APA ethical regulations for research on human subjects and all participants gave online informed consent, as approved by the institutional review boards of the principal investigators (Comité d'Éthique de la Recherche en Éducation et en Psychologie, CEREP-20-038-D). Questionnaire, raw data, code, and additional elements are publicly available on OSF at https://osf.io/t3xzy/.

### Participants and procedure

The total initial sample comprised 3617 Canadian participants at Wave 1 (April 2020). Participants were recruited by the polling firm Delvina from a representative web panel of more than one million Canadians. To ensure that participants took the survey seriously, responses of participants who took less than four minutes to complete a specific survey were excluded from the sample for that wave. Starting at Wave 2, we introduced two attention check items. Responses

**Table 1. Timetable for 10 waves of data collection from April 2020 to November 2020 for the full sample.**

| Waves | Starting date | Intervals | N | % Respondents | % Women | Mean age |
|-------|--------------|-----------|------|--------------|---------|----------|
| 1 | April 6 | 2 weeks | 3617 | 100% | 50.5% | 47.6 |
| 2 | April 21 | 2 weeks | 2282 | 63.0% | 49.0% | 49.0 |
| 3 | May 4 | 2 weeks | 2369 | 65.5% | 48.8% | 48.9 |
| 4 | May 18 | 2 weeks | 2296 | 63.5% | 48.5% | 48.9 |
| 5 | June 1 | 2 weeks | 2154 | 59.6% | 48.7% | 49.3 |
| 6 | June 15 | 2 weeks | 2116 | 58.5% | 48.8% | 49.4 |
| 7 | July 13 | 4 weeks | 2072 | 57.6% | 49.1% | 49.8 |
| 8 | August 17 | 5 weeks | 1871 | 51.7% | 49.4% | 50.4 |
| 9 | September 21 | 5 weeks | 1821 | 50.3% | 48.4% | 51.8 |
| 10 | November 26 | 9 weeks | 1883 | 52.5% | 48.4% | 50.3 |

*Note*: Percentages of respondents in each wave are calculated based on the Wave 1 sample.

of participants who failed both items in a specific wave were also excluded from the sample for that wave. Finally, to be consistent with the type of analysis used, participants who completed less than three waves of data collection were removed. Based on these exclusion criteria, the final sample included 2458 Canadians (1225 men and 1233 women). The sample was representative of the population in terms of gender, age, and province of residence. Age was measured as a continuous variable (see Table 1); 548 (22.3%) participants were 18–34 years old, 795 (32.3%) were 35–54, and 1115 (45.4%) were over 55.

We used longitudinal data collected in 10 waves over a period of eight months (see Table 1 for data collection timetable). In distributing the survey, we used a rolling cross-sectional (RCS) survey design [55]. For the first wave of our multi-wave study, RCS began with a large sample of respondents. The goal was to ensure that a minimum of 3500 Canadians completed the survey, involving approximatively 250 participants a day for 14 days.

For the second wave, the survey was sent to all participants exactly 14 days after they completed the first wave survey. We initially planned the COVID-19 survey to be administered in a total of 10 waves, for five months. However, because the COVID-19 pandemic persisted, we extended the study to cover a longer period of time. First, for Wave 1 (April 6, 2020) to Wave 6 (June 15, 2020), the COVID-19 survey was distributed every two weeks. Next, we prolonged the intervals between waves–Wave 7 (July 13, 2020) was distributed after four weeks, Wave 8 (August 17, 2020) after five weeks, and Wave 9 (September 21, 2020) after six weeks. Wave 10 started 10 weeks later (November 26, 2020). Participants were compensated for their time (for more information, see the technical report [56]).

## Planned missingness and missing values

To improve the validity of data collection, we used multi-form designs of planned missingness, allowing us to collect incomplete data from participants by arbitrarily assigning them to have missing items on a survey (for an overview, see [57]). We followed best practice procedures, using a variety of different versions of the same questionnaire. Each participant completed two-thirds of the total number of items. This multi-form design is useful when collecting data using numerous variables and when one has time limitations or concerns about respondent burden and fatigue [58, 59].

Additionally, as recommended in group-based trajectory modeling, we handled missing data (including planned missingness) via the full information maximum likelihood (FIML)

method. This allows for the inclusion of participants who have missing data on the dependent variables [60, 61].

## Demographic weighting

In order to work on a representative sample of the Canadian population, we relied on a design weight to correct identifiable demographic deviations from population characteristics [62]. The weighting process was conducted under the function "calibration" from the icarus package in R. We identified the best combination of calibration variables and fitting model by retaining the one that minimized the average estimation error on a range of 13 external benchmark measurements. These were based on data available from Statistics Canada. Calibrating with the "logit" method with respect to the variables such as the number of residents in the household, indigenous status, province of residence, age, and gender yielded the best results, with a bias reduction of 6.25%. The resulting weights ranged from 0.56 to 2.98 with a mean of 1.

## Measures

The questionnaire was divided into two sections: In the first section of Wave 1, the participants provided the main sociodemographic information. Specifically, they were asked to indicate their age, gender identity, province of residence, profession, education, and to answer COVID-19 related questions. In the second part of the questionnaire, participants responded to a series of psychological statements. Given that this was a large-scale longitudinal study, scales were shortened to include a greater number of constructs without overly taxing participants.

For the present study, the predictors (i.e., age, gender, and COVID-19 related questions) were collected at Wave 1 (April 6, 2020). We used items of compassionate self-responding from Wave 2 (April 21, 2020) to Wave 9 (September 21, 2020) and outcomes (emotional well-being, and sleep quantity and quality) from Wave 10 (November 26, 2020). Life satisfaction was measured at Wave 9 (September 21, 2020) and not at Wave 10. For each measure described below, participants rated the extent to which each statement applied to them using a 1 (*Not at all*) to 10 (*Completely*) response scale.

**COVID-19 related variables.** Participants were asked to answer two items assessing fear of infection ("How concerned are you about getting very sick with the virus") and fear for others ("How concerned are you about a loved one or friend getting very sick with the virus"). Fear of infection and fear for others were positively correlated ($r = .61$, $p < .001$) at Wave 1.

**Compassionate self-responding: Self-kindness, common humanity, and mindfulness.** We measured three positive facets of self-compassion in the COVID-19 context by adapting three items of the Self-Compassion Scale [22], one item per facet: (1) self-kindness ("When I don't like my own behavior during the current COVID-19 crisis, I try to be understanding and patient with myself"); (2) common humanity ("When I feel inadequate in my reaction to the current COVID-19 crisis, I try to remind myself that feelings of inadequacy are shared by most people"); and (3) mindfulness ("When something difficult happens to me related to the COVID-19 crisis, I try to take a balanced view of the situation"). Self-kindness was positively linked to common humanity ($r = .46$, $p < .001$) and mindfulness ($r = .44$, $p < .001$) and the two latter were positively correlated with each other ($r = .33$, $p < .001$) at Wave 2. Internal consistency ranged from acceptable to good (ω coefficient at T2 = .68; T3 = .76; T4 = .77; T5 = .75; T6 = .78; T7 = .78; T8 = .78; T9 = .80).

To assess the convergent validity of the three positive facets of self-compassion, we conducted an additional data collection (Wave 11, $N = 1671$). This data collection included the three adapted single items tailored to the COVID-19 context, as well as their respective single

**Table 2. Means and correlation matrix of all variables measured at Wave 11 ($N$ = 1671).**

|  | Mean (SD) | 1 | 2 | 3 | 4 | 5 | 6 |
|---|---|---|---|---|---|---|---|
| 1. Self-Kindness COVID-19 | 6.91 (2.03) | - | 0.69* | 0.67* | 0.56* | 0.50* | 0.48* |
| 2. Common Humanity COVID-19 | 6.76 (2.15) | - | - | 0.60* | 0.48* | 0.58* | 0.38* |
| 3. Mindfulness COVID-19 | 7.22 (1.95) | - | - | - | 0.51* | 0.41* | 0.53* |
| 4. Self-Kindness | 6.91 (1.95) | - | - | - | - | 0.59* | 0.62* |
| 5. Common Humanity | 6.71 (2.19) | - | - | - | - | - | 0.49* |
| 6. Mindfulness | 7.33 (1.92) | - | - | - | - | - | - |

*Note*:

* Correlation is significant at the .001 level.

items from the original scale. The correlation results presented in Table 2 demonstrated significant and strong associations between the COVID-19 items and the original items, supporting the convergent validity of our scale ($r_{self-kindness}$ = .56, $p < .001$, $r_{common\ humanity}$ = .58, $p < .001$, $r_{mindfulness}$ = .53, $p < .001$).

**Emotional well-being.** Emotional reactions during the COVID-19 pandemic were adapted from Reynolds et al. [63]. Participants read the following stem: "Everyone responds differently to the effects of a pandemic. This very difficult situation will impact different people in different ways. We would like to better understand how Canadians are reacting to the current state of the COVID-19 crisis" and then the participants rated 11 negative emotions (bored, helpless, lonely, angry, annoyed, afraid, frustrated, guilty, nervous, sad, worried, ω coefficient = .92) and one positive emotion (happy, $r$ = -.25, $p < .001$, when correlated with a composite score of negative emotions).

**Sleep quality and quantity.** We analyzed both the quantity and quality of sleep. First, we assessed the quantitative aspect of sleep by asking participants how much sleep, in hours and minutes, they got in the last 24 hours. Second, we measured the subjective qualitative aspect of sleep by asking participants to rate the quality of their sleep during the last 24 hours on a scale of 1 (*Slept very badly*) to 10 (*Slept very well*). Sleep quantity and quality were positively correlated ($r$ = .50, $p < .001$) at Wave 10.

**Life satisfaction.** Participants completed the Satisfaction With Life Scale (SWLS; [64]). The scale included five items (e.g., "I am satisfied with my life"), which demonstrated an excellent internal consistency (ω coefficient = .92).

## Statistical modeling

The statistical analyses and modeling were divided into three steps: (1) identifying multi-trajectory groups of compassionate self-responding (self-kindness, common humanity and mindfulness), (2) highlighting sociodemographic characteristics (i.e., gender and age) and COVID-19 related variables (e.g., fear of being infected) distinguishing groups of multi-trajectories and (3) analyzing the associations between trajectory group membership and five mental health indicators (i.e., negative emotions, happiness, sleep quantity and quality, and life satisfaction). Fig 1 summarizes the data analysis plan.

**Identifying multi-trajectory groups.** In step 1, we used a group-based method to identify the trajectories for self-kindness, common humanity, and mindfulness simultaneously ([60, 65]). Using finite mixtures of suitably defined probability distributions [61], the group-based approach for multinomial modeling of trajectories is a method that identifies and describes distinctive clusters of individual trajectories within the population. The multi-trajectory

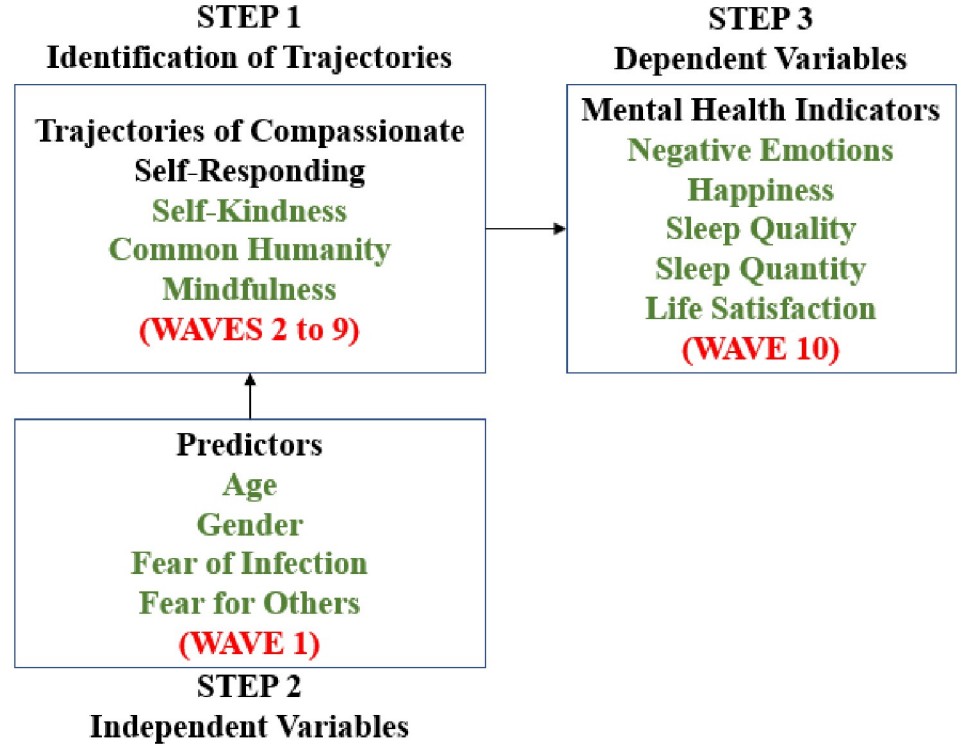

**Fig 1. Data analysis plan.**

approach identified groups of individuals that followed similar patterns across the three indicators of compassionate self-responding considered simultaneously in the same model [41].

The multi-trajectory approach is a generalization of group-based trajectory modeling, which allowed us to take full advantage of the information available in multivariate longitudinal data for tracking the course of self-compassion indicators over time [41]. In this context, we estimated latent classes from 1 to 4 and adjusted the shape of the trajectories, which can follow polynomial functions of time. In sum, multi-trajectory modeling allowed us to identify latent clusters of individuals following similar trajectories across multiple indicators (e.g., self-kindness, common humanity, and mindfulness) of a construct of interest (e.g., compassionate self-responding). These multi-trajectory group membership designations enabled us to create groups with different levels of overall compassionate self-responding (e.g., low, moderate, high) manifested during the COVID-19 period (Hypothesis 1).

A key issue in finite mixture modeling is determining the adequate number of trajectory groups and their shape for the best fitting model. Following the recommendations of D'Unger et al. [66], we used the Bayesian Information Criterion (BIC) and the Akaike information criterion (AIC) as a basis for selecting the optimal model, with BIC and AIC closer to zero indicating a better model fit. Based on the selected model, this procedure assigns people into trajectory groups based on the maximum posterior probability. Ultimately, because self-kindness, common humanity, and mindfulness were continuous with approximately normal distributions with some censoring at the extreme scores of the scale (i.e., 1 and 10), we modeled them using a censored normal distribution (CNORM).

Lastly, as the amount of time between measurement waves changed throughout the study (see Table 1), time was coded reflecting these variations in terms of number of weeks.

Specifically, Time 2 is coded 0, Time 3 is coded 2, Time 4 is coded 4, Time 5 is coded 6, Time 6 is coded 8, Time 7 is coded 10, Time 8 is coded 15, and Time 9 is coded 22.

**Associations between trajectory groups, sociodemographic factors, and COVID-19 related variables.** In step 2, we used the MULTRISK function of PROC TRAJ from SAS software to analyze sociodemographic risk factors (i.e., age and gender) and COVID-19 related variables (e.g., fear of being infected) to identify those that could differentiate compassionate self-responding multi-trajectory groups (Hypothesis 2).

**Associations between trajectory groups and mental health outcomes.** In step 3, using five independent univariate ANOVAs, we analyzed the relation between compassionate self-responding multi-trajectory groups and five mental health indicators measured 10 weeks after the main survey—negative emotions, happiness, sleep quantity and quality, and life satisfaction (Hypothesis 3). To compare group means, we relied on post-hoc comparisons using Tukey's HSD with Games Howell's correction.

## Results

### Multi-trajectory groups of self-kindness, common humanity, and mindfulness

To address the first goal, we identified the number and shape of the trajectories for the three positive indicators of self-compassion from Wave 2 to Wave 9, considered jointly in the same multi-trajectory model. Table 3 reports BIC and AIC scores for models with varying numbers of trajectory groups and trajectory shapes. Based on the BIC and AIC criteria, a four-group model was selected as the best fitting model.

Fig 2 shows the patterns of the trajectories in each indicator as well as the confidence intervals (i.e., 95% CIs). Interestingly, we observed that the four trajectories manifested similarly across indicators. In line with Hypothesis 1, participants experienced low ($N = 97$, 4.0% in purple), moderate-low ($N = 965$, 39.3% in orange), moderate-high ($N = 1149$, 46.7% in blue) and

**Table 3. Bayesian information criterion (BIC) and Akaike information criterion (AIC) for selection of models (Step 1).**

| Model | K | Indicator | Order | BIC | AIC |
|-------|---|-----------|-------|-----|-----|
| 1 | 1 | Self-kindness | 3 | -62142.46 | -62099.08 |
|   |   | Common humanity | 3 | | |
|   |   | Mindfulness | 3 | | |
| 2 | 2 | Self-kindness | 3, 3 | -59305.18 | -59224.21 |
|   |   | Common humanity | 3, 3 | | |
|   |   | Mindfulness | 3, 3 | | |
| 3 | 3 | Self-kindness | 3, 3, 3 | -58156.79 | -58038.23 |
|   |   | Common humanity | 3, 3, 3 | | |
|   |   | Mindfulness | 3, 3, 3 | | |
| 4 | 4 | Self-kindness | 3, 3, 3, 3 | -57716.98 | -57506.82 |
|   |   | Common humanity | 3, 3, 3, 3 | | |
|   |   | Mindfulness | 3, 3, 3, 3 | | |
| **5** | **4** | **Self-kindness** | **1, 0, 0, 0** | **-57636.28** | **-57569.77** |
|   |   | **Common humanity** | **0, 0, 1, 0** | | |
|   |   | **Mindfulness** | **0, 2, 1, 0** | | |

*Note*: K = Number of groups (trajectories). The order indicates whether the trajectory was fit with a constant (0), linear (1), quadratic (2), or cubic (3) function. For clarity, we only report the most relevant models (baseline model and best model for each k level).

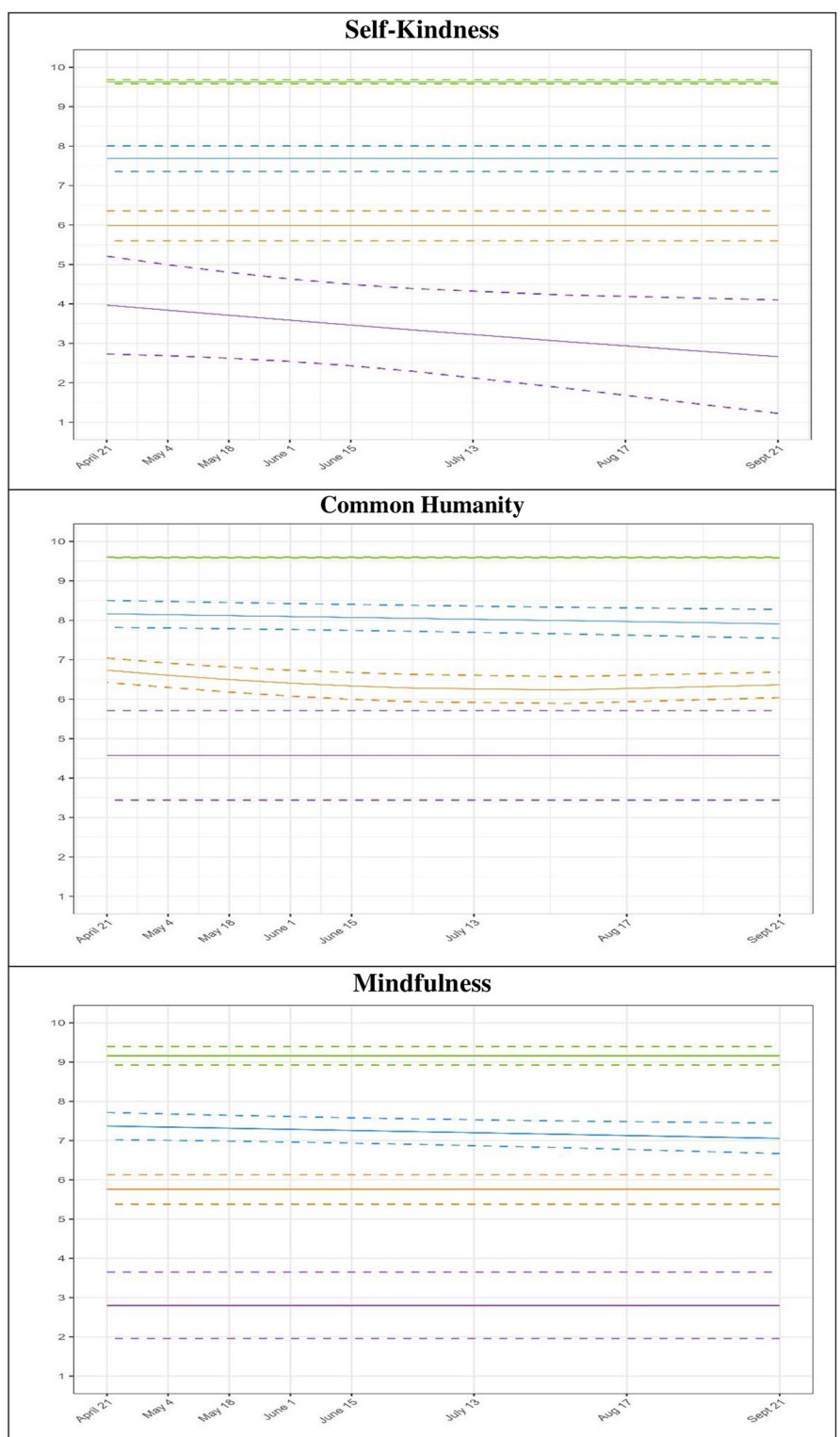

**Fig 2. Multi-trajectory groups with 95%CI (dashed lines) by dimensions.** Participants experiencing low ($N$ = 97, 4.0% in purple), moderate-low ($N$ = 965, 39.3% in orange), moderate-high ($N$ = 1149, 46.7% in blue), and high ($N$ = 247, 10.0% in green) levels across the three dimensions (from April 2020 to November 2020).

high ($N$ = 247, 10.0% in green) levels of compassionate self-responding. Specifically, two of the four trajectories showed moderate mean levels of compassionate self-responding (scores around 6 in the moderate-low trajectory group and scores around 8 in the moderate-high trajectory group, on a scale of 1–10), for each indicator. These two trajectories also tended to slightly decrease over time (i.e., negative linear and quadratic functions, see Table 3 and Fig 2).

Furthermore, and as expected for each indicator, a distinct cluster of individuals demonstrated a persistently high level of compassionate self-responding. This high trajectory group comprised of 10.0% of the sample, was characterized by a constant function, suggesting that these high trajectories were stable over time. Similarly, a cluster of individuals demonstrated a persistently low level of compassionate self-responding. This group included 4.0% of the sample and was characterized by a negative linear function for self-kindness and a constant function for common humanity and mindfulness. Taken together, these results indicated that distinct facets of compassionate self-responding were rather stable over time with minor fluctuations and diminutions for some individuals.

Fig 3 presents the daily new confirmed COVID-19 cases per million people in Canada [67]. As this figure shows, Canada experienced a peak in the number of confirmed death cases which started during the beginning of the data collection, around Wave 3 (which corresponded to the first "wave" of COVID-19). Interestingly, this peak led to an increase in COVID-19 sanitary measures (see [68, 69] but also [70, 71]) and corresponded to the decrease of compassionate self-responding trajectories, suggesting that this tremendous increased number of confirmed cases may have led to a decrease in self-kindness, common humanity, and mindfulness over time (from Wave 3 to Wave 9), for some individuals.

## Associations between trajectory groups, sociodemographic factors, and COVID-19 related variables

Our second goal was to identify sociodemographic factors (i.e., age and gender) and COVID-19 related variables (e.g., fear of being infected) that could differentiate these multi-trajectory

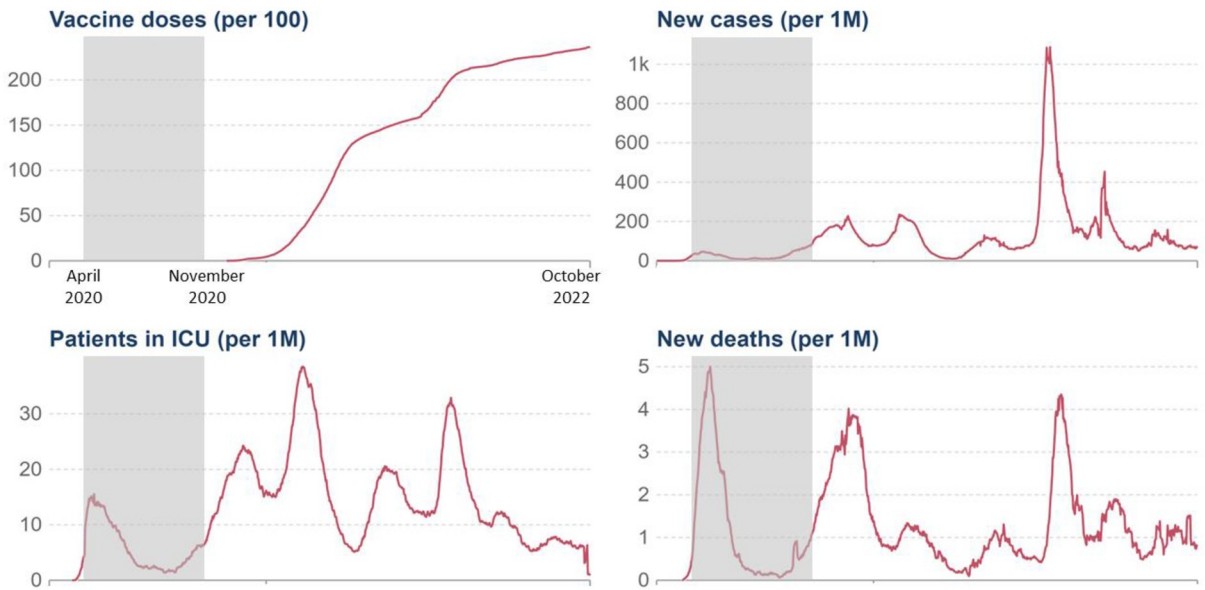

**Fig 3. Vaccine doses, daily new confirmed cases, ICU patients, and death cases in Canada.** *The grey zone indicates the period of our data collection.*

groups. Specifically, we aimed to determine whether the trajectory membership probabilities for the selected reference category (i.e., the trajectory group with low levels of compassionate self-responding) could be distinguished from other trajectory groups by considering sociodemographic and COVID-19 variables simultaneously.

Table 4 reports the factors associated with each trajectory group. For age, the larger the gap between trajectory groups (e.g., between low and moderate-low, low and high), the greater the absolute estimation (i.e., 0.15 from low vs. moderate-low and 0.48 for low vs. high). Consistent with Hypothesis 2, this significant and increased estimation suggests that older individuals were more likely to belong to the highest compassionate self-responding trajectory groups compared to younger individuals.

Furthermore, while women were slightly more likely to belong to high trajectories of compassionate self-responding as compared to men, these gender differences were not statistically significant. However, it is important to note that when gender is considered independently, we found significant and strong gender differences in favor of women. For a detailed analysis, please refer to the online supplementary material where we explored the interaction between age and gender. Specifically, older women had a higher probability of belonging to moderate-high and high trajectories, whereas older men had a higher probability of belonging to moderate-low and low trajectory groups.

Finally, and as expected, individuals who feared infection were more likely to belong to the highest compassionate self-responding trajectory groups. However, when the fear of infection extended to the family and friends, no difference was observed between trajectory groups.

## Associations between trajectory groups and mental health outcomes

To reach our third goal, we used five univariate ANOVAs to analyze the associations between multi-trajectory groups and negative emotions, happiness, sleep quantity, sleep quality, and life satisfaction measured 10 weeks after the main survey. Table 5 reports the means, standard deviations, $F$, degrees of freedom, $p$-values, effect sizes, and post-hoc Tukey's tests with Games Howell's correction for all variables as a function of trajectory groups. In line with Hypothesis 3, multi-trajectory groups reflecting higher levels of compassionate self-responding (vs. lower

**Table 4. Factors associated with trajectory groups (Step 2).**

|  | Parameter | Estimate | Standard error | T for H0 | Prob > \|T\| |
|---|---|---|---|---|---|
| Low (Reference category) | - | - | - | - | - |
| Low vs. Moderate Low | Constant | 1.05390 | 0.85743 | 1.229 | 0.2190 |
|  | Age | 0.15162 | 0.20514 | 0.739 | 0.4599 |
|  | Gender | -0.01356 | 0.55404 | -0.024 | 0.9805 |
|  | **Fear of infection** | **0.32009** | **0.09584** | **3.340** | **0.0008** |
|  | Fear for others | -0.10399 | 0.10075 | -1.032 | 0.3020 |
| Low vs. Moderate High | Constant | -0.69584 | 0.96371 | -0.722 | 0.4703 |
|  | **Age** | **0.44590** | **0.19604** | **2.275** | **0.0230** |
|  | Gender | -0.31228 | 0.52878 | -0.591 | 0.5548 |
|  | **Fear of infection** | **0.26591** | **0.10191** | **2.609** | **0.0091** |
|  | Fear for others | 0.04458 | 0.10876 | 0.410 | 0.6819 |
| Low vs. High | **Constant** | **-4.15005** | **1.44934** | **-2.863** | **0.0042** |
|  | **Age** | **0.48301** | **0.22609** | **2.136** | **0.0327** |
|  | Gender | -0.43107 | 0.58894 | -0.732 | 0.4642 |
|  | **Fear of infection** | **0.32323** | **0.11107** | **2.910** | **0.0036** |
|  | Fear for others | 0.21401 | 0.16973 | 1.261 | 0.2074 |

**Table 5. Means and standard deviations of all variables as a function of multi-trajectory groups (Step 3).**

|  | Negative emotions | Happiness | Sleep quality | Sleep quantity | Life satisfaction |
|---|---|---|---|---|---|
| Low | 4.96[a, b] (2.35) | 4.23[a] (3.13) | 6.29[a] (2.68) | 420[a] (105) | 5.05[a] (2.46) |
| Moderate-Low | 5.32[b] (1.93) | 5.19[a] (2.49) | 6.29[a] (2.27) | 426[a] (95) | 6.00[b] (1.96) |
| Moderate-High | 4.96[a, b] (2.14) | 5.67[b] (2.61) | 6.86[b] (2.30) | 437[a] (79) | 6.92[c] (1.86) |
| High | 4.84[a, b] (2.40) | 6.48[c] (3.18) | 7.46[c] (2.45) | 445[a] (93) | 7.23[c] (2.43) |
| $F$ | 4.47 | 11.66 | 9.74 | 2.32 | 44.45 |
| $df$ | 3, 1641 | 3, 1113 | 3, 1113 | 3, 1090 | 3, 1639 |
| $P$-value | .004 | < .001 | < .001 | .074 | < .001 |
| Partial $\eta^2$ | .023 | .048 | .053 | .007 | .074 |

*Note*: Standard deviations are presented in parentheses. Means with differing subscripts within columns are significantly different at the $p < .05$ using Tukey's HSD test with Games Howell's correction and refer to differences between trajectory groups. It is relevant to note that some non-significant mean differences might come from the fact that the group sizes of specific multi-trajectory groups are unequal and small, thus having an impact on the standard error.

levels) were associated with greater life satisfaction, more happiness, better sleep quality, greater sleep quantity (marginal), and lower degree of negative emotions. These results were robust when controlling for age and gender (except for the effect of trajectory groups on negative emotions, which became marginal).

## Exploratory analysis

To gain deeper and more nuanced understanding of the main variables of interest, we identified trajectories of self-kindness, common humanity, and mindfulness, separately. This allowed us to examine the co-occurrence of these distinct facets of compassionate self-responding. Table 6 shows the probabilities of co-occurrence of a specific trajectory group conditional to one of the two others. In line with Hypothesis 2, all trajectories were closely similar, suggesting that these three facets followed a comparable course over time. Specifically, the large diagonal elements of the probability matrices (in bold in Table 6) indicated that there was considerable overlap between these trajectories. These results indicated that the vast

**Table 6. Co-occurrence of self-kindness, common humanity, and mindfulness.**

| Group | Low | Moderate-low | Moderate-high | High |
|---|---|---|---|---|
| Probability of self-kindness group conditional on mindfulness group | | | | |
| High self-kindness | 0.0% | 0.0% | 2.6% | **71.3%** |
| Moderate-high self-kindness | 6.7% | 0.0% | **95.6%** | 23.7% |
| Moderate-low self-kindness | 4.8% | **97.2%** | 0.0% | 3.3% |
| Low self-kindness | **88.5%** | 2.8% | 1.8% | 1.8% |
| Probability of self-kindness group conditional on common humanity group | | | | |
| High self-kindness | 2.2% | 1.3% | 6.4% | **86.4%** |
| Moderate-high self-kindness | 21.5% | 0.0% | **90.3%** | 13.6% |
| Moderate-low self-kindness | 17.5% | **96.0%** | 3.0% | 0.0% |
| Low self-kindness | **59.1%** | 2.7% | 0.3% | 0.0% |
| Probability of common humanity group conditional on mindfulness group | | | | |
| High common humanity | 7.9% | 0.0% | 2.3% | **45.0%** |
| Moderate-high common humanity | 0.0% | 4.5% | **73.0%** | 34.2% |
| Moderate-low common humanity | 19.9% | **90.2%** | 22.0% | 15.4% |
| Low common humanity | **72.2%** | 5.4% | 2.8% | 5.4% |

majority of individuals who followed the highest level of self-kindness trajectory also followed the highest common humanity (86.4%) and mindfulness (71.3%) trajectories. We observed similar patterns for the moderate-high, moderate-low, and low trajectory. For example, participants who followed a low trajectory on one indicator were more likely to follow the same trajectory on both other indicators.

## Discussion

The main goal of the present paper was to investigate the positive facets of self-compassion as key factors that could help people better cope with the psychological challenges in the context of the COVID-19 pandemic. Specifically, we aimed at better understanding the heterogeneity in trajectories of self-kindness, common humanity, and mindfulness during the COVID-19 pandemic. Moreover, although numerous studies have already explored the detrimental psychological consequences of the COVID-19 pandemic, little is known regarding factors that can buffer against challenges and adversity brought on by the COVID-19 and their negative effects on mental health. First, in line with our hypotheses, we identified clusters of individuals following persistently low, moderate-low, moderate-high, and high trajectories of self-compassion, across three facets of compassionate self-responding, with differences in their stability over time. Second, we observed several gender, age and COVID-19-related differences. Third, we demonstrated that trajectory groups reflecting higher levels of compassionate self-responding were associated with better mental health 10 weeks after the main survey.

### Self-kindness, common humanity, and mindfulness over time

The results of this research evidenced four distinct trajectories that demonstrated variations between individuals in their levels of self-kindness, common humanity, and mindfulness. While most individuals aggregated between moderate-low (39.3%) and moderate high levels (46.7%), a substantial number of people reported low (4.0%) and high (10.0%) levels over time, which can be seen as an approximation of a latent normal distribution of compassionate self-responding scores over time. This is consistent with studies showing population heterogeneity in people's tendency to love, accept, criticize, and castigate themselves [72].

Interestingly, we found that distinct facets of compassionate self-responding were rather stable over time (i.e., constant shape) with minor fluctuations and diminutions for some groups (i.e., linear and quadratic forms). Importantly, this decrease in self-kindness, common humanity, and mindfulness over time (from Wave 3 to Wave 9) among some individuals is remarkably associated with the peak in new cases and death experience in Canada during this period (see Fig 3), suggesting that this highly stressful period might have had a negative impact on the way people perceived and experienced the difficulties related to the pandemic.

On the one hand, we found strong inter-individual differences in compassionate self-responding scores which remained quite stable over time, suggesting that these positive facets of self-compassion manifested as a trait. On the other hand, the greater variation observed for some individuals (often those with lower levels of self-compassion) suggested that compassionate self-responding may, to some extent, have also manifested as a state, influenced by the global context as well as daily routines and life events. Consistent with these findings, research has demonstrated that though self-compassion is a relatively stable trait-like characteristic, it is nevertheless a modifiable skill that can be enhanced using interventions promoting self-compassion (e.g., [73, 74]).

## Age, gender, and COVID-19-related differences in self-compassion

The present study evidenced that older people have a higher probability to belong to high trajectory groups. These results are consistent with the literature suggesting that older people have higher levels of self-compassion [45]. Specifically, self-compassion is linked with greater fulfillment of potential and can serve as an asset in resolving the developmental tasks of late life (e.g., loss of loved ones, declines in health and function). Rather than responding to these undesired life changes with anger and self-criticism, a self-compassionate mindset enables individuals to cope with the challenges of aging by treating themselves with kindness and love, regarding their circumstances as common to humanity, and maintaining an objective and balanced perspective on negative emotions [75]. Because the COVID-19 context created similar threats (e.g., fear infection, health anxiety), we expected our results to echo this literature, showing that older adults were more likely to report compassionate self-responding, to cope with life changing experiences during the pandemic, as compared to young people.

Interestingly, and contrary to our hypothesis, we found that older women (55 years-old and older) reported greater compassionate self-responding as compared to the other age and gender groups. A meta-analysis regarding gender differences in self-compassion [51] showed an opposite pattern with men having slightly higher scores than women. However, Yarnell et al. [52] argued that gender differences in self-compassion should be nuanced as gender socialization norms may be putting some persons at a disadvantage in terms of adopting this adaptive way of coping with difficult emotions and life events. Ultimately, although we found some age and gender differences, this longitudinal study adds to the growing evidence that compassionate self-responding, and by extension self-compassion, is an asset for psychological flourishing for both men and women at any age.

To go further, our study took a closer look at the impact of fear of infection on compassionate self-responding trajectories. Interestingly, we observed a distinction between personal fear of infection and fear extended to family members and friends. While personal fear was linked to higher compassionate self-responding, the extension of fear to others seemed to negate this effect. This suggests that facing the direct threat of COVID-19 prompted individuals to adopt a more self-compassionate stance, possibly as a coping mechanism in the face of adversity. The lack of significant association between fear extended to family members and compassionate self-responding trajectories suggests that while personal fear may lead to greater self-compassion, fear for other people may generate broader compassion rather than compassion for oneself.

## The role of self-compassion in mental health

One of the main goals of this study was to examine the association between compassionate self-responding and indicators related to mental health. In line with previous findings (e.g., [76, 77]), we demonstrated that trajectory groups reflecting higher levels of compassionate self-responding were associated with greater life satisfaction, more happiness, better sleep quality, higher sleep quantity (marginal), and lower levels of negative emotions, as compared to trajectory groups reflecting lower levels of compassionate self-responding. This study replicates well-established findings in the literature suggesting that self-compassion is positively associated with mental health (e.g., [78, 79]). These results suggest that positive facets of self-compassion may attenuate the negative consequences of the COVID-19 pandemic, for example, through greater kindness towards oneself. In this vein and in line with previous findings (e.g., [80–84]), future research should continue to explore whether self-compassion-based as well as mindfulness-based interventions (e.g., yoga, meditation training, learning self-compassion skills, school programs) that foster kindness, compassion, and acceptance toward the self,

may have a positive impact on mental health of young and older men and women, as well as of people experiencing adverse events.

## Limitations

Although this study expands our understanding of how compassionate self-responding manifests over time, this study is not without limitations. While the original Self-Compassion Scale [22] contains 26 items, including positive and negative statements, we relied on a three-item version with one positive item per facet selected, which we then adapted to the context of the pandemic to specifically investigate compassionate self-responding in the context of COVID-19-related events. Even though we had successfully assessed convergent validity of our items in the additional 11[th] wave of data collection, these three selected items may not have sufficiently captured self-kindness, common humanity, and mindfulness during the pandemic for all individuals. In other words, some individuals may have varied to a greater extent on facets of self-compassion that we did not measure and thus, the validity of the compassionate self-responding items may have been reduced for some participants. Future work should explore all facets of self-kindness, common humanity, and mindfulness, as well as their counterpart (i.e., self-judgment, isolation, and over-identification) in different contexts to fully capture self-compassion manifestations.

## Implications and future directions

Research examining mental health during the COVID-19 pandemic from a self-compassionate perspective is flourishing. Associations found in the present study enrich the current literature and suggest implications for research and practice.

**Research.**   Most psychological methodologies focus on interindividual variations between individuals at one time point (e.g., mean, correlation) which inadequately capture interindividual variations in intraindividual change over time. The person-oriented approach explains the interindividual variations in intraindividual change (see [85–87]) and provides unique insight to detect developments or changes in the characteristics of the target population over time. This increases the validity of the data and makes it easier to find long-term patterns. Specifically, the present research relied on a multi-trajectory latent class growth analysis (LCGA) for identifying homogeneous subgroups within the larger heterogeneous population [40]. Following this longitudinal approach, trajectories were intentionally designed to span time 2 to time 9, encompassing predictors from time 1 and evaluating outcome variables at time 10, two months post-survey. This strategic approach aimed to establish preliminary evidence for causal inferences while mitigating overlaps [41]. Doing so, we provided convincing elements in favor of the predicting role of self-compassion in mental health over time. To go further, a complementary approach would be to explore joint trajectories or incorporate time-varying covariates to gain a comprehensive perspective on temporal dynamics between compassionate self-responding trajectories and mental health indicators.

**Practice.**   Scholars have demonstrated the positive effect of self-compassion and mindfulness specifically on mental health ([80, 88]). Adding to these findings, the results of our study may help individuals improve their quality of life, especially during the context of global pandemics and other dramatic social changes. Importantly, the greater variation observed for some individuals suggests that compassionate self-responding may fluctuate over time and thus, be modified by specific factors. Consequently, practitioners such as therapists, counselors, activists, social policy makers, teachers, instructors, and organizational administrators could include self-compassion interventions within their training and mentoring efforts. The current study contributes to practice by identifying self-compassion as a potential key factor

associated with mental health that practitioners could exploit to develop and deliver preventive actions and effective interventions aimed at supporting people in times of crises [89].

## Conclusion

The current study demonstrates promising associations between self-kindness, common humanity, mindfulness, and indicators related to mental health during the COVID-19 crisis. Specifically, individuals high in compassionate self-responding were more likely to report greater emotional well-being and life satisfaction, better sleep quality, and greater sleep quantity. This suggests that self-compassion may be a buffer against adversity, allowing for greater resilience in times of crises.

## Author Contributions

**Conceptualization:** Robin Wollast, Éric Lacourse, Geneviève A. Mageau, Véronique Dupéré, Roxane de la Sablonnière.

**Data curation:** Robin Wollast, Mathieu Pelletier-Dumas, Anna Dorfman.

**Formal analysis:** Robin Wollast, Éric Lacourse, Mathieu Pelletier-Dumas.

**Funding acquisition:** Roxane de la Sablonnière.

**Methodology:** Robin Wollast, Éric Lacourse, Mathieu Pelletier-Dumas, Roxane de la Sablonnière.

**Project administration:** Roxane de la Sablonnière.

**Supervision:** Roxane de la Sablonnière.

**Writing – original draft:** Robin Wollast.

**Writing – review & editing:** Robin Wollast, Éric Lacourse, Geneviève A. Mageau, Mathieu Pelletier-Dumas, Anna Dorfman, Véronique Dupéré, Jean-Marc Lina, Dietlind Stolle, Roxane de la Sablonnière.

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
