## [Decision Letter · Decision Letter 0]

24 Jun 2022

PONE-D-22-01485The dynamics of self-compassion in Canadian adults during the first 8 months of the COVID-19 pandemic: A person-oriented multi-trajectory approachPLOS ONE

Dear Dr. Wollast,

Thank you for submitting your manuscript to PLOS ONE. After careful consideration, we feel that it has merit but does not fully meet PLOS ONE’s publication criteria as it currently stands. Therefore, we invite you to submit a revised version of the manuscript that addresses the points raised during the review process.

The manuscript has been evaluated by two reviewers, and their comments are available below.

The reviewers have raised a number of major concerns. They request improvements to the reporting of methodological aspects of the study, for example to clarify how self-compassion has been operationalised and  the associations between the latent classes and covariates/outcomes.

Could you please carefully revise the manuscript to address all comments raised?

We look forward to receiving your revised manuscript.

Kind regards,

Lorena Verduci

Staff Editor

PLOS ONE

Journal Requirements:

This study is part of a larger national research project financed by the Canadian Institutes of Health Research (CIHR) that examines the social consequences of the COVID-19 pandemic on Canadians (for more information, see https://csdc-cecd.wixsite.com/covid19csi?lang=en)

This study is part of a larger national research project financed by the Canadian Institutes of Health Research (CIHR) that examines the social consequences of the COVID-19 pandemic on Canadians

However, funding information should not appear in the Acknowledgments section or other areas of your manuscript. We will only publish funding information present in the Funding Statement section of the online submission form. 

This study is part of a larger national research project financed by the Canadian Institutes of Health Research (CIHR) that examines the social consequences of the COVID-19 pandemic on Canadians (for more information, see https://csdc-cecd.wixsite.com/covid19csi?lang=en)

5. We note you have included a table to which you do not refer in the text of your manuscript. Please ensure that you refer to Table 6 in your text; if accepted, production will need this reference to link the reader to the Table.

Reviewers' comments:

Reviewer's Responses to Questions

**Comments to the Author**

1. Is the manuscript technically sound, and do the data support the conclusions?

Reviewer #1: Partly

Reviewer #2: Partly

2. Has the statistical analysis been performed appropriately and rigorously? 

Reviewer #1: I Don't Know

Reviewer #2: Yes

3. Have the authors made all data underlying the findings in their manuscript fully available?

Reviewer #1: Yes

Reviewer #2: Yes

4. Is the manuscript presented in an intelligible fashion and written in standard English?

Reviewer #1: Yes

Reviewer #2: Yes

5. Review Comments to the Author

Reviewer #1: Also uploaded in a word document.

An interesting study with some good findings, just potentially not framed in the best way. In general, the paper could be re-written more concisely, with more targeted references and justification. There are some methodological concerns on how self-compassion has been operationalised, with limited evidence that the adapted measure has been tested for validity. I do feel that some more effort is needed to highlight the relevance of the work, how it will add to previous literature, and more work is needed to clearly present the results. The paper is framed as looking at the relationship between self-compassion and mental health over the COVID-19 pandemic, however, only positive elements of self-compassion, and outcomes associated with mental health are measured. Based on this I feel like the justification for the study set out in the introduction is not coherent with the study conducted. In the results, it is not clear how the trajectories of self-compassion changed over the pandemic. This could be presented more clearly so a person with minimal knowledge of trajectory modelling could understand where there were fluctuations in self-compassion over time, in which groups, and by how much. Results do indicate that those with high self-compassion also had better outcomes in factors associated with mental health, however, mental health was only recorded once so cause and effect is difficult to ascertain, which is a shame due to the longitudinal design.

More specific comments are written below.

Introduction

In general, the introduction is very long and needs to be made more concise. There is limited justification for why only compassionate responding is used and why factors associated with mental health outcomes are looked at rather than direct measures of mental health etc.

1) Neff (2003) is referenced for the conceptualisation of self-compassion and justification for the research, but the method does not measure self-compassion. It only measures levels of compassionate responding and ignores levels of uncompassionate responding. However, according to Neff, self-compassion consists of 3 sets of opposing elements to form a global self-compassion score (self-kindness vs. self-judgement, mindfulness vs. over identification and common humanity vs isolation). The introduction discusses the opposing elements of compassionate responding (e.g. self-judgement, isolation and over-identification) to form the justification for this research, yet these elements are not measured. There needs to be some justification in the introduction why this study only looks at the three positive elements of self-compassion.

2) The language used to describe self-compassion also needs changing to reference it as compassionate self-responding rather than self-compassion, as it currently implies that global self-compassion is being measured, e.g. :

“In addition, while it has been demonstrated that the combination of self-kindness, common humanity, and mindfulness reflects a global score of self-compassion (Neff et al., 2017)”

“…how each dimension of self-compassion changes over time…”

3) The majority of the research referenced in relation to mental health also uses the self-compassion scale which measures all 6 elements, not just 3. The evidence in relation to mental health needs to be more in line with the compassionate responding, rather than global self-compassion, to strengthen the argument.

4) The introduction is very focused on self-compassions relationship with mental health directly, including levels of depression, anxiety etc. But the study methods do not look at mental health directly, it looks at outcomes related to mental health (negative emotions, happiness, sleep quantity, sleep quality, and life satisfaction). The introduction should be tailored to discuss the link between self-compassion and those specific outcomes, rather than leaning so heavily on mental health specifically.

5) In general, the Self-Compassion and Sociodemographic Factors section would benefit from being reworded. It is quite confusing, and the justification for looking at age reads that younger people have worse mental health and therefore must have worse self-compassion, when there are lots of variables and factors that impact mental health in younger people differently. It would be better to take a softer approach and state that it is unknown whether self-compassion differs across age groups, which this study will explore.

6) “In the same vein,” is used repeatedly in the document. I suggest using different wording to improve readability.

7) I think the introduction would benefit from breaking the hypothesis and goals down into research questions. The goals of the study are clear, but it is unclear what questions will be answered if the goals are achieved.

Method

1) Demographic information would be better presented as a table. Age could be presented as a mean and range rather than percentage in age categories.

2) It would be good to include the year when the dates of the waves are given. The pandemic spanned multiple years so it is unclear whether the data was collected early in the pandemic.

3) Planned missingness – this section is quite passive. It is unclear how missingness was dealt with. It is mentioned that multiple versions of the questionnaire were used but it is unclear what this means? Do the questions change, the order, the wording?

4) The measures would benefit from some indices of validity and reliability.

5) I have some reservations about the self-compassion measures that were used.

i. One aim of the study was to “examine the developmental trajectories of the three self-compassion dimensions during the COVID-19 crisis”. However, by adapting the wording of the measure what you are actually measuring is the level of compassionate self-responding in direct relation to suffering caused directly by COVID-19, not just general self-compassion levels “during” the COVID-19 crisis.

ii. I raise this because the wording of the self-compassion items are difficult to interpret and I imagine participants interpretation of “related to the COVID-19 crisis” element of the items varied quite widely. Did you do any qualitative work with participants to explore their understanding of the question? If so, it would be good to include this to justify the wording.

iii. Did you run any validity checks comparing responses to these items to items from the self-compassion scale?

iv. Do you have a measure of reliability of these singular items?

b. Usually standardised measures of mental health ask respondents about their mood or behaviour over a period of time. However, the measure of mood used here is a state-based measure rating current levels of particular emotions. This limits what conclusions can be drawn about mental health. For example, the person may have been really calm when filling in the questionnaire, but over the past two weeks had been highly worried.

Results

- Multinomial modeling of trajectories is not something I have read about before, so any comments come from the perspective of a lay reader.

o I was expecting to see some analysis that demonstrated the degree of change in self-compassion over time from baseline, for each trajectory group.

o It is mentioned that there are groups that show more persistent patterns, but there is no evidence that the change in pattern within the trajectory groups were significantly different between groups. It is unclear how the “more persistent pattern” was identified.

o It is unclear from the results reported and the figures what the overall trajectories of self-compassion look like. It would be good to include a figure of each respondents overall self-compassion scores over time.

- The exploratory analysis does not seem necessary. It is not in the remit of the paper, and opens up a completely new goal, and set of interpretations that have not been discussed in the introduction. I would recommend removing this section.

Discussion

1) There are times that new information is built into the discussion that has not been mentioned in the rest of the paper. For example:

a. Strikingly, this peak corresponds to the shape of self-compassion trajectories, suggesting that this tremendous increased number of confirmed cases may have led to a decrease in self-kindness, common humanity and mindfulness over time (from wave 3 to wave 9), among individuals belonging to low trajectory groups – this is not presented in the results section and is not displayed in any of the figures for the reader to confirm. Putting this in the results section in a figure to show the corresponding change in self-compassion would be beneficial.

b. The discussion about self-compassion’s structure. This could be used in the intro as a justification for focusing on three components, but it does not fit well in the discussion.

2) Conclusions are drawn which suggest self-compassion is a stable trait and also a state, however from my interpretation of the results, there is limited evidence that self-compassion changes drastically. It seems from Figure 2 that respondents levels of self-compassion stay in the remit of their trajectory group. Therefore, this suggests that perhaps self-compassion levels may change slightly over time and circumstance, but that in general, the categorical level of self-compassion will stay the same, therefore adding more support for self-compassion as a trait then a state.

3) The paper is also limited by common method bias which can inflate correlation scores.

4) I would also suggest that paper is limited by measures of mental health outcomes only being measured at one time point. There is evidence that mental health outcomes fluctuated as self-compassion levels fluctuated.

Other

a. Figure 1 needs more detail to inform us about the data analysis plan. It does not inform us about IV/DV, excluded data, missing data, etc…

Reviewer #2: The authors should be commended with respect to the clarity of their writing; it is clear that a lot of work has gone into the proofreading and editing of this document before submission, which has made the review process much easier. In terms of the technical processes taken, a lot of what the authors have conducted is on the right track, therefore I will just describe the parts of their methodology and conclusions that may benefit from a second pass. I had a few queries about the specifics of the analysis for the authors to consider, which may have an impact on their choice of model and the conclusions drawn.

1) The authors seem to rely on BIC and BIC alone when making decisions on the best-fitting model to the data ("Based

on the BIC criterion, a four-group model was selected as the best fitting model."). Mixture models have an array of model adequacy criteria to peruse as the investigator, however. I would appreciate it if the authors not only presented their BIC-based conclusions, but also considered the array of additional model adequacy criteria regarding the count of classes. As outlined in excellent papers that address this issue specifically such as van der Nest et al. (2020; https://doi.org/10.1016/j.alcr.2019.100323), other coefficients such as the Vuong-Lo-Mendell-Rubin Likelihood Ratio Test are available to help inform decisions related to the count of classes that is optimal. Noting that the authors are using PROC TRAJ in SAS, and as noted in van der Nest et al. the fit coefficients are similar to Stata in being very limited when informing this part of model selection, I am curious if the authors can describe other aspects of class enumeration that they considered? For example, Nylund-Gibson and Choi's (2018; http://dx.doi.org/10.1037/tps0000176) general tutorial article on class enumeration in an LCA context describes considerations such as the minimum proportion of participants estimated to be members of a class in order to help with establishing class stability. They provide an example of asking whether a small proportion of the sample captured by a class is like to be stable, using a rough rule of thumb of approximately 5% as something large enough. Alternatively, Muthen's presentations on mixture models provide examples of valid latent classes with <5% of the sample as being valid. I think the authors need to flesh out their model selection considerations beyond that of just the BIC coefficient, keeping in mind the kinds of considerations discussed prior, to show some additional nuance in their decision making prior to interpretation. Employing the D’Unger et al. (1998) resource to inform model selection from over two decades ago isn't adequate given the evolution of mixture modelling in the meantime.

2) Can the authors cite the basis of their interpretation of the posterior assignment probabilities as being adequate when over >.70? The van der Nest article I've cited prior might help them do this, specifically the section on APPA. Currently the values are presented without justification.

3) When looking at the associations between the estimated (and I would emphasise the latter term) class assignments and sociodemographic covariates / distal outcomes, the authors seem to be employing a one-step approach. Much of the recent literature on covariate / distal outcome prediction involving mixture models has heavily cautioned against employing this approach. The authors should refer to the summary provided by Asparouhov and Muthén (2014; https://doi.org/10.1080/10705511.2014.915181) for a description of the shortcomings of the one-step approach, which have been addressed by the contemporary three-step approaches to covariate and distal outcome estimation. If the three-step BCH approach for distal outcome estimation (i.e., for the mental health outcomes based on latent class), or the three-step covariate estimation approach (i.e., for sociodemographic variables) are not available in SAS, then the authors will need to concretely defend the validity of their proposed associations between the latent classes and covariates/outcomes.

4) The final sentence of the Results is a bit of a mystery - what did you find? Was it the same when MI was employed? How many imputations did you run, etc.?

5) Referring to non-linear trajectories as an 'unstable shape' is tricky within the context of mixture models. This may be interpreted as the scaled entropy of the trajectory class becoming poorer over time, for example. Consider referring to these trajectories as non-linear.

6) When viewing the plots of the trajectories, their non-linearity seems minor. Furthermore, the generally flat trajectories of the classes makes me wonder how the LCGA models compare with the much simpler growth models that don't split the trajectories within the sample. Acknowledging that the red class (3.2% of the sample) appears to be the lone class that isn't flat in trajectory shape, and is relatively marginal in the count of participants classified in this class, can the authors compare their findings with a generic growth model that will likely turn out to be relatively flat across the waves of measurement? There's a parsimony consideration here, and while person-centered analyses provide the opportunity for additional detail about these trajectories, the value of these distinct classes versus being able to describe people scoring high/med-high etc. over time based on a lone flat trajectory is worth considering in the Discussion.

I hope these comments/queries have been useful in helping with your publication, and thank you for the opportunity to provide reviewer comments.

6. PLOS authors have the option to publish the peer review history of their article (what does this mean?). If published, this will include your full peer review and any attached files.

Reviewer #1: No

Reviewer #2: No

---

## [Author Response · Author response to Decision Letter 0]

17 Oct 2022

See Word document (Response to Reviewers). Thank you.

---

## [Decision Letter · Decision Letter 1]

4 Dec 2022

PONE-D-22-01485R1

Compassionate self-responding in Canadian adults during the first 8 months of the COVID-19 pandemic: A person-oriented multi-trajectory approach

PLOS ONE

Dear Dr. Wollast,

Thank you for submitting your manuscript to PLOS ONE. After careful consideration, we have decided that your manuscript does not meet our criteria for publication and must therefore be rejected.

I am sorry that we cannot be more positive on this occasion, but hope that you appreciate the reasons for this decision.

Kind regards,

Ali B. Mahmoud, Ph.D.

Academic Editor

PLOS ONE

Reviewers' comments:

Reviewer's Responses to Questions

**Comments to the Author**

1. If the authors have adequately addressed your comments raised in a previous round of review and you feel that this manuscript is now acceptable for publication, you may indicate that here to bypass the “Comments to the Author” section, enter your conflict of interest statement in the “Confidential to Editor” section, and submit your "Accept" recommendation.

Reviewer #1: (No Response)

2. Is the manuscript technically sound, and do the data support the conclusions?

Reviewer #1: No

3. Has the statistical analysis been performed appropriately and rigorously? 

Reviewer #1: No

4. Have the authors made all data underlying the findings in their manuscript fully available?

Reviewer #1: Yes

5. Is the manuscript presented in an intelligible fashion and written in standard English?

Reviewer #1: Yes

6. Review Comments to the Author

Reviewer #1: I have uploaded my comments in a word document. Also copied below.

I don’t believe many of my initial review points have been taken on board or edited. I have included examples of this below. As these points have not been addressed, either through edits in the document or through responses from the authors with defence as to why these decisions were made I want to recommend that this paper is not published. I do not trust that these authors have a full grasp of what self-compassion is and have not put thought behind the operationalisation of the concepts of self-compassion or mental health and therefore cannot justify why they have used them. If a lay person was to read this paper, they would learning inaccurate information about what self-compassion is and how it is defined. There has been no effort to rectify this from when I initially reviewed the paper.

The above is particularly important because the authors have edited the self-compassion measure and have only measured the three elements. As the self-compassion scale is designed to measure all 6 facets with specific wording, I am unconfident on the validity of this measure but some justification in the introduction may have helped alongside some pilot analysis showing that the measure is associated with outcomes from the SCS - but I am still very sceptical about the validity and reliability of this measure

In turn I have concerns about have conclusions drawn on how self-compassion protected mental health during the pandemic. Self-compassion was measured across 10 waves, but we have no clear indication of attrition here, and mental health was recorded at one time point. An improved study would have measured mental health at each time point and looked to understand how mental health fluctuated with self-compassion.

Point 1: Introduction

Comments from my first review:

“1) Neff (2003) is referenced for the conceptualisation of self-compassion and justification for the research, but the method does not measure self-compassion. It only measures levels of compassionate responding and ignores levels of uncompassionate responding. However, according to Neff, self-compassion consists of 3 sets of opposing elements to form a global self-compassion score (self-kindness vs. self-judgement, mindfulness vs. over identification and common humanity vs isolation). The introduction discusses the opposing elements of compassionate responding (e.g. self-judgement, isolation and over-identification) to form the justification for this research, yet these elements are not measured. There needs to be some justification in the introduction why this study only looks at the three positive elements of self-compassion.”

2) The language used to describe self-compassion also needs changing to reference it as compassionate self-responding rather than self-compassion, as it currently implies that global self-compassion is being measured, e.g. :

“In addition, while it has been demonstrated that the combination of self-kindness, common humanity, and mindfulness reflects a global score of self-compassion (Neff et al., 2017)”

These points just have not been addressed. The same text is included in the introduction and no effort has been applied to justify their choices for only focusing on the positive elements of self-compassion. As such, I have grave concerns about the theoretical and conceptual understanding of the work. The definition provided for self-compassion is inaccurate, stating that there are only 3 facets of self-compassion, and not six as clearly conceptualised in Neff 2003b. This reference has been used to support the definition provided in this paper and it is inaccurate. If the paper wanted to focus on the positive facets of self-compassion, this would be fine and Neff (2018) states that people can focus on either compassionate responding or uncompassionate responding independently if this fits the aim of their work, however this paper has not justified why they are only focusing on positive elements and have not considered the implications of this at any point in their paper. This is poor scientific practice. If people were to read this article alone without understanding of self-compassion, their teachings would be inaccurate and biased to the author’s cause. As this comment hasn’t been addressed, and the definition of self-compassion has just been changed to remove the negative elements this suggests the authors have no justification for not including the negative elements which makes me concerned about their understanding and interpretation of the findings.

Only including the positive elements of self-compassion makes the data vulnerable to common method bias and inflated ITT.

Methodology, Results and Discussion

It is still unclear how low, moderate (high & low) and high self-compassion were determined. What were the cut off points? How were these decided. This still hasn’t been addressed from my first review. As the measure is not in line with the validated measure of self-compassion I would like to fully understand how cut offs were determined and how this linked to the cut offs associated with the validated self-compassion scale.

There is no discussion of how assumptions were checked.

As there was a high attrition rate, how was this dealt with in the analysis?

Other comments not addressed from first review:

I would have expected these points to be added to the limitations section:

3) The paper is also limited by common method bias which can inflate correlation scores – particularly as only positive elements of the SCS were used.

4) I would also suggest that paper is limited by measures of mental health outcomes only being measured at one time point. There is evidence that mental health outcomes fluctuated as self-compassion levels fluctuated.

7. PLOS authors have the option to publish the peer review history of their article (what does this mean?). If published, this will include your full peer review and any attached files.

Reviewer #1: No

- - - - -

---

## [Author Response · Author response to Decision Letter 1]

2 Jan 2023

Please find all of our comments in the PDF file titled "Response to Reviewers"

---

## [Decision Letter · Decision Letter 2]

27 Apr 2023

PONE-D-22-01485R2

Compassionate self-responding in Canadian adults during the first 8 months of the COVID-19 pandemic: A person-oriented multi-trajectory approach

PLOS ONE

Dear Dr. Wollast,

Thank you for submitting your manuscript to PLOS ONE. After careful consideration, we feel that it has merit but does not fully meet PLOS ONE’s publication criteria as it currently stands. Therefore, we invite you to submit a revised version of the manuscript that addresses the points raised during the review process.

We look forward to receiving your revised manuscript.

Kind regards,

Giuseppe Marano

Academic Editor

PLOS ONE

Journal Requirements:

2. In the Methods section please include the informed consent statement to reflect whether "written or verbal" informed consent was obtained from all participants for inclusion in the study.

Additional Editor Comments (if provided):

Reviewers' comments:

Reviewer's Responses to Questions

**Comments to the Author**

1. If the authors have adequately addressed your comments raised in a previous round of review and you feel that this manuscript is now acceptable for publication, you may indicate that here to bypass the “Comments to the Author” section, enter your conflict of interest statement in the “Confidential to Editor” section, and submit your "Accept" recommendation.

Reviewer #3: All comments have been addressed

Reviewer #4: (No Response)

Reviewer #5: (No Response)

Reviewer #6: (No Response)

Reviewer #7: (No Response)

2. Is the manuscript technically sound, and do the data support the conclusions?

Reviewer #3: Yes

Reviewer #4: Partly

Reviewer #5: Partly

Reviewer #6: (No Response)

Reviewer #7: Partly

3. Has the statistical analysis been performed appropriately and rigorously? 

Reviewer #3: Yes

Reviewer #4: I Don't Know

Reviewer #5: I Don't Know

Reviewer #6: (No Response)

Reviewer #7: Yes

4. Have the authors made all data underlying the findings in their manuscript fully available?

Reviewer #3: Yes

Reviewer #4: Yes

Reviewer #5: Yes

Reviewer #6: (No Response)

Reviewer #7: Yes

5. Is the manuscript presented in an intelligible fashion and written in standard English?

Reviewer #3: Yes

Reviewer #4: Yes

Reviewer #5: Yes

Reviewer #6: (No Response)

Reviewer #7: Yes

6. Review Comments to the Author

Reviewer #3: The authors have thoroughly addressed all the edits requested by previous reviewers. The paper is methodologically sound and provides an interesting addition to the literature. I suggest this paper is accepted with no further revisions.

Reviewer #4: Using a longitudinal research design, the authors aim to identify and describe the individual trajectories of self-kindness, common humanity, and mindfulness during the first eight months of the COVID-19 pandemic. The paper is interesting in its objective, but I have some concerns about the method chosen to evaluate the constructs and the general organization of the paper. In addition, some comments will be mainly directed at improving the organization of the paper, which appears difficult to follow and, in some pieces, redundant and not flowing.

The introduction section appears complicated to follow, with continuous references to the study's objectives and hypotheses that should be clarified once with the support of the scientific literature.

The references to mental health during the covid era are referred mainly to 2020, and some more recent papers should be presented, as much literature has been produced.

The authors should reorganize the introductory section by clarifying the constructs investigated and the relationship between them and then subsequently define objectives and hypotheses in a clear and defined way and consequently define the identified method. I suggest integrating the paragraph Compassionate Self-Responding and Sociodemographic Factors.

The reading and fluency of the paper can benefit from this change.

Regarding the method section, I am uncertain about the authors' choice of using Reynolds' adaptation for emotion analysis. Could the authors clarify the choice regarding the use of this tool? The authors talk about mental health, but they assessed the degree experienced in certain emotions, which does not measure mental health. I suggest the concept of well-being instead.

Define whether "gender" refers to "sex assigned at birth" or "gender identity."

My primary concern is about the validity and reliability of the Compassionate Self-Responding measure because authors refer to "dimensions" but have only measured three elements (items). this aspect needs to be stressed more.

I suggest summarizing the part explaining the analyses because it is verbose.

The discussions are centered on the results obtained however need to fully clarify the limitation found in the method.

Reviewer #5: This paper from Canada explores trajectories of compassionate self-responding (CSR) during the COVID19 pandemic using 10 waves of data collection in a longitudinal research design. In addition it aims to compare this data with socio demographic variables and mental health indicators. The study team have done an excellent work producing an important study finding 4 clusters of individual trajectories of CSR. I feel this is an important addition into the body of literature in positive psychology especially in the context of the COVID19 pandemic.

The claims made in the paper with respect to the different trajectories and their correlation with gender and age seem sound and deserves publications. However, I feel the paper will benefit by addressing the following comments prior to publication.

1. One of the 3 purposes of the research is to look at correlation between the trajectories of CSR and mental health indicators. This is linked to Hypothesis 3 “We hypothesize that high multi- trajectory groups will be positively associated with indicators of good mental health (negative emotions, happiness, sleep quality and quantity, and life satisfaction)” However, despite it being a longitudinal study they have collected indicators associated with mental health only at the end of the study, namely, participants’ negative affect, happiness, life satisfaction, sleep quantity and quality. Further, some variables such as sleep quality and quantity capture only that of the last 24 hours which is likely to give limited information. It is unclear why mental health variables were collected only in the last wave of the study despite the longitudinal design. Hence, I feel the purpose and hypothesis 3 and the associated conclusions cannot be fairly made and the phrasing of associated claims is not fully supported by the data and analyses. Although this is explicitly stated as a limitation, unless there is further evidence, I feel the claims under purpose and hypothesis 3 may be modified.

2. The authors describe CSR as falling under the global construct of self-compassion and having 3 interconnected dimensions –positive and negative. The authors also quote Neff, 2003b, saying the original Self-Compassion Scale has 26 items. The study uses only 3 positive dimensions and of the 26 items authors choose only 3 items. The justification for this is in the Limitations as “This was based on the highest factor loadings from the original validation study which we then adapted to the context of the pandemic”. This justification needs to be moved to earlier and expanded further. It is also unclear whether this new scale was validated.

3. It is unclear why they omitted all negative items. This is despite the authors stating that “However, these studies focused on the combined effects of compassionate self-responding and its negative counterparts (i.e., self-judgment, isolation, and over-identification) such that it is not clear if these associations were due to the positive or rather the negative dimensions of self-compassion”. Based on this argument it would have made sense to add some of the negative items as well. This needs to be explained.

4. The study claims to have looked at ‘context’. In addition there is multiple mentions of ‘COVID19 events’ (e.g.: Intro says “Accordingly, we focused on compassionate self- responding while facing COVID-19 events”. ) However, it’s unclear why they have not included any variables related to COVID-19 events in the survey (such as COVID19 infection of self, COVID19 infections of loved ones and news of death). The claims regarding the correlations seem to be based solely on Figure 3 where they have looked at 'daily new confirmed COVID-19 cases per million people in Canada'. Hence, I feel claims related to any association between the self-compassionate responding and COVID19 related events need to be modified to clarify that the correlations between COVID19 and self-compassionate responding is based on the timing of data collection, national level daily new confirmed COVID-19 cases and not specific individual COVID19 related events.

Overall this is an important study that brings useful insights about trajectories of SCR. Hence, I hope these comments are useful to refine the paper so that the important findings about the clusters of individual trajectories of SCR and their correlation with age and gender are shared in a fair manner. Thank you for this opportunity.

Reviewer #6: (No Response)

Reviewer #7: Thank you for the opportunity to review this paper. I think it is an important area of research in order to discover positive key factors that help people better cope with the psychological challenges associated with the COVID-19 pandemic.

However, there are a number of edits that I recommend be made before a possible publication. They are highlighted below in the order in which they appear.

- Verbal tenses should be revised all over the manuscript and conjugate in the past;

- In introduction authors wrote “in this challenging time” or “current pandemic”, I suggest to refer to the past;

- I suggest to move the aims reported in the first section of introduction (“The purpose … social change.”) to the end and under the subheading “Overview and hypothesis”, summarize the section e aims and changing the subheading title in – i.e. – “Aims and hypothesis”; moreover, there is a part that is a result: “in a large representative sample of Canadians (N = 2458 from an initial sample size of 3617 participants). ”

- In introduction, talking about gender, authors write “This may explain by the fact that women…during lockdown”: it is a strong sentence and despite sharable authors should provide some evidence to support;

- In method and measures “Questionnaire can be found in supplementary materials” is repeated;

- In results, the part: “To test this, we used the chi-square goodness-of-fit test including its adjusted standardized residuals to determine whether the multi-trajectory membership probabilities were higher for women (vs. men) and higher for younger adults (v. older adults). Specifically, cross-tabulations between four latent classes of compassionate self-responding trajectories and sociodemographic characteristics (i.e., age and gender) were conducted.” And “To reach our third goal, we used five univariate ANOVA to analyze the associations between multi-trajectory groups and negative emotions, happiness, sleep quantity and quality, and life satisfaction.” Is method, move to statistical modelling in method;

- In discussion: “The general research question of the present paper was to investigate positive key factors that help people better cope with the psychological challenges associated with the COVID-19 pandemic.” Is not corresponding to the three objectives and hypothesis.

- In discussion several sentences are too strong, such as “…compassionate self-responding were associated with greater life satisfaction…”: I suggest to use conditional using for example “more likely to be associated…);

- In introduction authors cite “preventive intervention”, but none of them or future prospective are proposed in discussion section;

- In discussion the phrase “the greater variation observed for some individuals suggests that compassionate self-responding can also manifest as a state influenced by the global context as well as daily routines and life events.” Iss too strong;

- The citation Limcaoco refers to a paper in medRxiv in 2020. I think a paper not published in a peer-review journal should not be cited, especially if two-three years has passed;

- Considering methods, authors stated that self-compassion scale is designed with 6 items, whereas in this study 3 items were used: I asking if the validity is guaranteed and I am not so confident. I appreciated the fact that authors discussed this particular point, justifying it with the interviewer engagement, but I think this aspect should be highlighted in method and discussion.

Although this comments, it is evident that authors did a lot of work in the writing and proofreading of the manuscript.

Thank you again for allowing me to review your paper. I hope you will consider these revisions, as I think this paper does provide some really valuable information addressing an unmet need to understand the wellbeing of population and varibales that can act on negative effects of COVID-19 pandemic on mental health..

7. PLOS authors have the option to publish the peer review history of their article (what does this mean?). If published, this will include your full peer review and any attached files.

Reviewer #3: No

Reviewer #4: No

Reviewer #5: No

Reviewer #6: No

Reviewer #7: No

---

## [Author Response · Author response to Decision Letter 2]

29 Aug 2023

A PDF version of this letter is available.

Reviewers' comments:

Reviewer's Responses to Questions

Comments to the Author

1. If the authors have adequately addressed your comments raised in a previous round of review and you feel that this manuscript is now acceptable for publication, you may indicate that here to bypass the “Comments to the Author” section, enter your conflict of interest statement in the “Confidential to Editor” section, and submit your "Accept" recommendation.

Reviewer #3: All comments have been addressed

Reviewer #4: (No Response)

Reviewer #5: (No Response)

Reviewer #6: (No Response)

Reviewer #7: (No Response)

2. Is the manuscript technically sound, and do the data support the conclusions?

Reviewer #3: Yes

Reviewer #4: Partly

Reviewer #5: Partly

Reviewer #6: (No Response)

Reviewer #7: Partly

3. Has the statistical analysis been performed appropriately and rigorously?

Reviewer #3: Yes

Reviewer #4: I Don't Know

Reviewer #5: I Don't Know

Reviewer #6: (No Response)

Reviewer #7: Yes

4. Have the authors made all data underlying the findings in their manuscript fully available?

Reviewer #3: Yes

Reviewer #4: Yes

Reviewer #5: Yes

Reviewer #6: (No Response)

Reviewer #7: Yes

5. Is the manuscript presented in an intelligible fashion and written in standard English?

Reviewer #3: Yes

Reviewer #4: Yes

Reviewer #5: Yes

Reviewer #6: (No Response)

Reviewer #7: Yes

 

6. Review Comments to the Author

Reviewer #3: The authors have thoroughly addressed all the edits requested by previous reviewers. The paper is methodologically sound and provides an interesting addition to the literature. I suggest this paper is accepted with no further revisions.

Response: We would like to thank reviewer #3 for his/her positive and encouraging feedback on our revised manuscript. 

Reviewer #4: Using a longitudinal research design, the authors aim to identify and describe the individual trajectories of self-kindness, common humanity, and mindfulness during the first eight months of the COVID-19 pandemic. The paper is interesting in its objective, but I have some concerns about the method chosen to evaluate the constructs and the general organization of the paper. In addition, some comments will be mainly directed at improving the organization of the paper, which appears difficult to follow and, in some pieces, redundant and not flowing.

Response: We would like to thank reviewer #4 for his/her constructive and valuable comments on our work. 

The introduction section appears complicated to follow, with continuous references to the study's objectives and hypotheses that should be clarified once with the support of the scientific literature.

Response: Based on the reviewer’s comments, we have improved and streamlined the introduction. For instance, we have included newer references, rewritten the mental health section, renamed some subheadings, and provided additional modifications. Below, we detailed all of these changes. We hope that this newly improved version is clearer for you and for future readers.

The references to mental health during the covid era are referred mainly to 2020, and some more recent papers should be presented, as much literature has been produced.

Response: As suggested, we have included more recent references and several meta-analyses in the mental health section. Specifically, we wrote (page 3): “The COVID-19 pandemic and its consequences have negatively affected mental health worldwide. In a cross-national study involving 48 countries, Gamonal-Limcaoco et al. (2022) found high levels of negative emotions in the general population, with higher levels among women, young adults and those who expressed concern about getting infected. Consistent with these findings, several other studies have documented the negative impact of COVID-19 on emotional experiences. Based on the pooled effect sizes of 19 studies, Ernst et al. (2022) found an overall small increase in loneliness since the start of the pandemic (see also Buecker & Horstmann, 2021 for similar results). Similarly, in their systematic review and meta-analysis, Salari et al. (2020) highlighted the prevalence of stress, anxiety, and depression among the general population during the COVID-19 pandemic. In the same vein, two meta-analyses demonstrated that the COVID-19 pandemic increased negative emotions and disrupted sleep quality (Jahrami et al., 2021; Oliveira Carvalho et al., 2021). Moreover, Satici et al. (2020) found that the fear of COVID-19 was positively associated with psychological distress and negatively associated with life satisfaction. Other studies have demonstrated an increase in psychological and mental health disorders globally during the COVID-19 pandemic (Leung et al., 2022).”

The authors should reorganize the introductory section by clarifying the constructs investigated and the relationship between them and then subsequently define objectives and hypotheses in a clear and defined way and consequently define the identified method. I suggest integrating the paragraph Compassionate Self-Responding and Sociodemographic Factors. The reading and fluency of the paper can benefit from this change.

Response: Thank you for this suggestion. We have integrated the “Compassionate Self-Responding and Sociodemographic Factors” in the introduction to facilitate the reading. We believe this change indeed enhances fluency. 

Regarding the method section, I am uncertain about the authors' choice of using Reynolds' adaptation for emotion analysis. Could the authors clarify the choice regarding the use of this tool? The authors talk about mental health, but they assessed the degree experienced in certain emotions, which does not measure mental health. I suggest the concept of well-being instead.

Response: Thank you for bringing this error to our attention. We indeed relied on the items used by Reynolds and colleagues (2007). However, we cited the wrong paper in the reference section, which may have caused confusion. We have now cited the appropriate reference (see below). In this context, we measured emotional well-being, assessed via positive and negative emotions, as a common indicator of mental health. 

Reynolds, D. L., Garay, J. R., Deamond, S. L., Moran, M. K., Gold, W., & Styra, R. (2007). Understanding, compliance and psychological impact of the SARS quarantine experience. Epidemiology & Infection, 136(7), 997-1007. https://doi.org/10.1017/S0950268807009156

Define whether "gender" refers to "sex assigned at birth" or "gender identity."

Response: In the measures section, we now write: “In the first section of Wave 1, the participants provided the main sociodemographic information. Specifically, they were asked to indicate their age, gender identity, province of residence, profession, education, and to answer COVID-19 related questions.”

My primary concern is about the validity and reliability of the Compassionate Self-Responding measure because authors refer to "dimensions" but have only measured three elements (items). this aspect needs to be stressed more.

Response: We thank you for this comment and the opportunity to clarify and enhance our analysis. First, we now clarify that each dimension of self-responding was assessed with a single item in the “Introduction” and “Method” sections. Additionally, we are now referring to “facet” instead of “dimension” throughout the manuscript. Most importantly, we have collected additional data to test for their convergent validity. Specifically, we wrote (page 8): “To assess the convergent validity of the three positive facets of self-compassion, we conducted an additional data collection (Wave 11, N = 1671). This data collection included the three adapted single items tailored to the COVID-19 context, as well as their respective single items from the original scale. The correlation results presented in Table 2 demonstrated significant and strong associations between the COVID-19 items and the original items, supporting the convergent validity of our scale (rself-kindness= .56, p < .001, rcommon humanity = .58, p < .001, rmindfulness = .53, p < .001).”

I suggest summarizing the part explaining the analyses because it is verbose.

Response: We agree with you that the “Statistical Modeling” section is complex. Indeed, during the first round of revision, some reviewers requested more detailed information about the analysis strategy which led us to expand this part, in order to provide as much information as possible for scholars who wish to replicate our findings using similar methods. However, we agree with your suggestion and have trimmed (a little bit) parts when possible. 

The discussions are centered on the results obtained however need to fully clarify the limitation found in the method.

Response: Thank you for this important comment. We have created a limitations section in which we carefully detailed several limitations including (a) operationalization of the scale, (b) assessment of mental health, and (c) correlational design. We believe that these points address the main limitations of the present work. If you believe that other elements should be included, we will be happy to do so.

Thank you again for your careful revision.

Reviewer #5: This paper from Canada explores trajectories of compassionate self-responding (CSR) during the COVID19 pandemic using 10 waves of data collection in a longitudinal research design. In addition it aims to compare this data with socio demographic variables and mental health indicators. The study team have done an excellent work producing an important study finding 4 clusters of individual trajectories of CSR. I feel this is an important addition into the body of literature in positive psychology especially in the context of the COVID19 pandemic. The claims made in the paper with respect to the different trajectories and their correlation with gender and age seem sound and deserves publications. However, I feel the paper will benefit by addressing the following comments prior to publication.

Response: We would like to thank reviewer #5 for his/her very detailed and useful revision of the manuscript.

1. One of the 3 purposes of the research is to look at correlation between the trajectories of CSR and mental health indicators. This is linked to Hypothesis 3 “We hypothesize that high multi-trajectory groups will be positively associated with indicators of good mental health (negative emotions, happiness, sleep quality and quantity, and life satisfaction)” However, despite it being a longitudinal study they have collected indicators associated with mental health only at the end of the study, namely, participants’ negative affect, happiness, life satisfaction, sleep quantity and quality. Further, some variables such as sleep quality and quantity capture only that of the last 24 hours which is likely to give limited information. It is unclear why mental health variables were collected only in the last wave of the study despite the longitudinal design. Hence, I feel the purpose and hypothesis 3 and the associated conclusions cannot be fairly made and the phrasing of associated claims is not fully supported by the data and analyses. Although this is explicitly stated as a limitation, unless there is further evidence, I feel the claims under purpose and hypothesis 3 may be modified.

Response: We agree that this is a good point to consider, and in designing the study we opted for the approach we presented in our paper. In the present study, trajectories were intentionally designed to span time 2 to time 9, encompassing predictors from time 1 and evaluating outcome variables at time 10, two months post-survey. This strategic approach aimed to establish preliminary evidence for causal inferences while mitigating overlaps (Nagin et al., 2018). It provides answers to a specific research question “Do self-compassion trajectories predict mental health indicators?” As you pointed out, another research question that would be important to examine is whether trajectories of self-compassion vary jointly with trajectories of mental health indicators. Future studies could be designed to explore joint trajectories or incorporate time-varying covariates to gain a comprehensive perspective on temporal dynamics between compassionate self-responding trajectories and mental health indicators. We have made this change in the discussion (see pp. 16-17).

2. The authors describe CSR as falling under the global construct of self-compassion and having 3 interconnected dimensions –positive and negative. The authors also quote Neff, 2003b, saying the original Self-Compassion Scale has 26 items. The study uses only 3 positive dimensions and of the 26 items authors choose only 3 items. The justification for this is in the Limitations as “This was based on the highest factor loadings from the original validation study which we then adapted to the context of the pandemic”. This justification needs to be moved to earlier and expanded further. It is also unclear whether this new scale was validated.

Response: We have clarified this in the introduction (see pages 3 and 4). Most importantly, we have collected additional data to test for their convergent validity. Specifically, we wrote: “To assess the convergent validity of the three positive facets of self-compassion, we conducted an additional data collection (Wave 11, N = 1671). This data collection included the three adapted single items tailored to the COVID-19 context, as well as their respective single items from the original scale. The correlation results presented in Table 2 demonstrated significant and strong associations between the COVID-19 items and the original items, supporting the convergent validity of our scale (rself-kindness= .56, p < .001, rcommon humanity = .58, p < .001, rmindfulness = .53, p < .001).”

Table 2. Means and correlation matrix of all variables measured at Wave 11 (N = 1671)

 Mean (SD) 1 2 3 4 5 6

1. Self-Kindness COVID-19 6.91 (2.03) - 0.69* 0.67* 0.56* 0.50* 0.48*

2. Common Humanity COVID-19 6.76 (2.15) - - 0.60* 0.48* 0.58* 0.38*

3. Mindfulness COVID-19 7.22 (1.95) - - - 0.51* 0.41* 0.53*

4. Self-Kindness 6.91 (1.95) - - - - 0.59* 0.62*

5. Common Humanity 6.71 (2.19) - - - - - 0.49*

6. Mindfulness 7.33 (1.92) - - - - - -

Note: * Correlation is significant at the .001 level.

We are also attaching a Word version of this letter to make it easier to read this new table.

3. It is unclear why they omitted all negative items. This is despite the authors stating that “However, these studies focused on the combined effects of compassionate self-responding and its negative counterparts (i.e., self-judgment, isolation, and over-identification) such that it is not clear if these associations were due to the positive or rather the negative dimensions of self-compassion”. Based on this argument it would have made sense to add some of the negative items as well. This needs to be explained.

Response: We agree that including the negative aspects of self-compassion would have been interesting. We had decided to focus on positive aspects of self-compassion during the COVID-19 pandemic because positive aspects that foster kindness, acceptance, and affection toward the self tend to play a major role in self-compassion interventions (Egan et al., 2022). This justification is presented on page 4. We also clarified our argument on pages 3-4 to focus solely on compassionate self-responding.

4. The study claims to have looked at ‘context’. In addition there is multiple mentions of ‘COVID19 events’ (e.g.: Intro says “Accordingly, we focused on compassionate self- responding while facing COVID-19 events”.) However, it’s unclear why they have not included any variables related to COVID-19 events in the survey (such as COVID19 infection of self, COVID19 infections of loved ones and news of death). The claims regarding the correlations seem to be based solely on Figure 3 where they have looked at 'daily new confirmed COVID-19 cases per million people in Canada'. Hence, I feel claims related to any association between the self-compassionate responding and COVID19 related events need to be modified to clarify that the correlations between COVID19 and self-compassionate responding is based on the timing of data collection, national level daily new confirmed COVID-19 cases and not specific individual COVID19 related events.

Response: As suggested, we have modified the word “events” when necessary and improved our analysis by including COVID-19 related variables. In this context, we used a new analysis strategy and reported it within the manuscript. Specifically:

“COVID-19 related variables. Participants were asked to answer two items assessing fear of infection (“How concerned are you about getting very sick with the virus”) and fear for others (“How concerned are you about a loved one or friend getting very sick with the virus”). Fear of infection and fear for others were positively correlated (r = .61, p < .001) at Wave 1.” Note that we initially included infection status but given that only 5 persons got infected in the whole sample, we decided to omit this variable from the analysis.

“Associations between Trajectory Groups, Sociodemographic Factors, and COVID-19 Related Variables. In step 2, we used the MULTRISK function of PROC TRAJ from SAS software to analyze sociodemographic risk factors (i.e., age and gender) and COVID-19 related variables (e.g., fear of being infected) to identify those that could differentiate compassionate self-responding multi-trajectory groups (Hypothesis 2).”

“Finally, individuals who feared infection were more likely to belong to the highest compassionate self-responding trajectory groups. Interestingly, when the fear of infection extended to the family and friends, no difference was observed between trajectory groups.” 

Table 4. Factors associated with trajectory groups (Step 2)

 Parameter Estimate Standard error T for H0 Prob > |T|

Low (Reference category) - - - - -

Low vs. Moderate Low Constant 1.05390 0.85743 1.229 0.2190

 Age 0.15162 0.20514 0.739 0.4599

 Gender -0.01356 0.55404 -0.024 0.9805

 Fear of infection 0.32009 0.09584 3.340 0.0008

 Fear for family -0.10399 0.10075 -1.032 0.3020

Low vs. Moderate High Constant -0.69584 0.96371 -0.722 0.4703

 Age 0.44590 0.19604 2.275 0.0230

 Gender -0.31228 0.52878 -0.591 0.5548

 Fear of infection 0.26591 0.10191 2.609 0.0091

 Fear for family 0.04458 0.10876 0.410 0.6819

Low vs. High Constant -4.15005 1.44934 -2.863 0.0042

 Age 0.48301 0.22609 2.136 0.0327

 Gender -0.43107 0.58894 -0.732 0.4642

 Fear of infection 0.32323 0.11107 2.910 0.0036

 Fear for family 0.21401 0.16973 1.261 0.2074

In the discussion, we wrote this: “To go further, our study took a closer look at the impact of fear of infection on compassionate self-responding trajectories. Interestingly, we observed a distinction between personal fear of infection and fear extended to family members and friends. While personal fear was linked to higher compassionate self-responding, the extension of fear to others seemed to negate this effect. This suggests that facing the direct threat of COVID-19 prompted individuals to adopt a more self-compassionate stance, possibly as a coping mechanism in the face of adversity. The lack of significant association between fear extended to family members and compassionate self-responding trajectories suggests that while personal fear may lead to greater self-compassion, fear for other people may generate broader compassion rather than compassion for oneself.”

Ultimately, we have modified Figure 1 accordingly.

Overall this is an important study that brings useful insights about trajectories of SCR. Hence, I hope these comments are useful to refine the paper so that the important findings about the clusters of individual trajectories of SCR and their correlation with age and gender are shared in a fair manner. Thank you for this opportunity.

Response: Thank you very much for your encouraging words and valuable input about our work. We believe our work has greatly improved and we are looking forward to reading your new feedback on this revised version.

Reviewer #6: (No Response)

Reviewer #7: Thank you for the opportunity to review this paper. I think it is an important area of research in order to discover positive key factors that help people better cope with the psychological challenges associated with the COVID-19 pandemic. However, there are a number of edits that I recommend be made before a possible publication. They are highlighted below in the order in which they appear.

Response: We thank reviewer #7 for his/her pertinent and helpful comments on the manuscript.

- Verbal tenses should be revised all over the manuscript and conjugate in the past. In introduction authors wrote “in this challenging time” or “current pandemic”, I suggest to refer to the past;

Response: Thank you for this useful comment. The manuscript was edited for English language usage and grammar by a native English-speaking proofreader who reviewed the entire manuscript. We hope the result will meet your expectations.

- I suggest to move the aims reported in the first section of introduction (“The purpose … social change.”) to the end and under the subheading “Overview and hypothesis”, summarize the section e aims and changing the subheading title in – i.e. – “Aims and hypothesis”; moreover, there is a part that is a result: “in a large representative sample of Canadians (N = 2458 from an initial sample size of 3617 participants). ”

Response: Thank you for this thoughtful suggestion. We have renamed the section “Overview of the Present Study and Hypotheses” with “Aims and Hypotheses”. After careful consideration, we decided to keep a short summary at the beginning of the introduction but removed the part you suggested from the paragraph.

- In introduction, talking about gender, authors write “This may explain by the fact that women…during lockdown”: it is a strong sentence and despite sharable authors should provide some evidence to support;

Response: We have slightly rephrased this sentence to make it less direct. Importantly, we have now cited the work from Carli (2020) investigating the associations between gender inequality during the COVID-19 pandemic.

Carli, L. L. (2020). Women, gender equality and COVID-19. Gender in Management: An International Journal, 35(7/8), 647-655. https://doi.org/10.1108/GM-07-2020-0236

- In method and measures “Questionnaire can be found in supplementary materials” is repeated;

Response: Thank you for highlighting this redundancy. We have removed one of them.

- In results, the part: “To test this, we used the chi-square goodness-of-fit test including its adjusted standardized residuals to determine whether the multi-trajectory membership probabilities were higher for women (vs. men) and higher for younger adults (v. older adults). Specifically, cross-tabulations between four latent classes of compassionate self-responding trajectories and sociodemographic characteristics (i.e., age and gender) were conducted.” And “To reach our third goal, we used five univariate ANOVA to analyze the associations between multi-trajectory groups and negative emotions, happiness, sleep quantity and quality, and life satisfaction.” Is method, move to statistical modelling in method;

Response: We decided to change our analysis strategy for this analysis. We now rely on the MULTRISK function from PROC TRAJ in SAS software. This approach is more robust to capture longitudinal data and we successfully replicated our results with these new advanced techniques. Having said that, we have moved the methodological aspect of the analysis strategy in the method section to make it clear for the reader. Thank you for pointing this out.

- In discussion: “The general research question of the present paper was to investigate positive key factors that help people better cope with the psychological challenges associated with the COVID-19 pandemic.” Is not corresponding to the three objectives and hypothesis.

Response: Thank you for pointing out this imprecision. We have modified the sentence accordingly: “The main goal of the present paper was to investigate the positive facets of self-compassion as key factors that could help people better cope with the psychological challenges in the context of the COVID-19 pandemic.” 

We then described in the same paragraph the three objectives and hypotheses.

- In discussion several sentences are too strong, such as “…compassionate self-responding were associated with greater life satisfaction…”: I suggest to use conditional using for example “more likely to be associated…);

Response: We were very careful not to include any causal language in the manuscript and have modified some sentences as you suggested to reflect the fact that observed associations may not replicate in another sample.

- In introduction authors cite “preventive intervention”, but none of them or future prospective are proposed in discussion section;

Response: Our results provided preliminary evidence supporting that self-compassion-based trainings can help people in difficult times. We developed this approach in the discussion section as an efficient intervention strategy. Following your suggestion, we have included a sentence in the subsection “Practice” in the “Implications and Future Directions” of the discussion section: “The current study contributes to practice by identifying self-compassion as a potential key factor associated with mental health that practitioners could exploit to develop and deliver preventive actions and effective interventions aimed at supporting people in times of crises (MacDonald & Neville, 2023).”

MacDonald, H. Z., & Neville, T. (2023). Promoting college students’ mindfulness, mental health, and self-compassion in the time of COVID-19: feasibility and efficacy of an online, interactive mindfulness-based stress reduction randomized trial. Journal of College Student Psychotherapy, 37(3), 260-278. https://doi.org/10.1080/87568225.2022.2028329

- In discussion the phrase “the greater variation observed for some individuals suggests that compassionate self-responding can also manifest as a state influenced by the global context as well as daily routines and life events.” Is too strong;

Response: As you suggested, we tempered this sentence: “On the other hand, the greater variation observed for some individuals suggested that compassionate self-responding may, to some extent, have also manifested as a state, influenced by the global context as well as daily routines and life events.”

- The citation Limcaoco refers to a paper in medRxiv in 2020. I think a paper not published in a peer-review journal should not be cited, especially if two-three years has passed;

Response: Thank you for pointing out this incorrect reference. We have included the published one and adjusted the text accordingly: “In a cross-national study involving 48 countries, Gamonal-Limcaoco et al. (2022) found high levels of negative emotions in the general population, with higher levels among women, young adults and those who expressed concern about getting infected.”

Gamonal-Limcaoco, S., Montero-Mateos, E., Lozano-López, M. T., Maciá-Casas, A., Matías-Fernández, J., & Roncero, C. (2022). Perceived stress in different countries at the beginning of the coronavirus pandemic. The International Journal of Psychiatry in Medicine, 57(4), 309-322. https://doi.org/10.1177/00912174211033710

- Considering methods, authors stated that self-compassion scale is designed with 6 items, whereas in this study 3 items were used: I asking if the validity is guaranteed and I am not so confident. I appreciated the fact that authors discussed this particular point, justifying it with the interviewer engagement, but I think this aspect should be highlighted in method and discussion.

Response: We agree with your comment and believe this is a good point to consider. Following your suggestion, we have collected additional data to test for their convergent validity (see new Table 2). Specifically, we wrote: “To assess the convergent validity of the three positive facets of self-compassion, we conducted an additional data collection (Wave 11, N = 1671). This data collection included the three adapted single items tailored to the COVID-19 context, as well as their respective single items from the original scale. The correlation results presented in Table 2 demonstrated significant and strong associations between the COVID-19 items and the original items, supporting the convergent validity of our scale (rself-kindness= .56, p < .001, rcommon humanity = .58, p < .001, rmindfulness = .53, p < .001).”

Although this comments, it is evident that authors did a lot of work in the writing and proofreading of the manuscript. Thank you again for allowing me to review your paper. I hope you will consider these revisions, as I think this paper does provide some really valuable information addressing an unmet need to understand the wellbeing of population and variables that can act on negative effects of COVID-19 pandemic on mental health.

Response: Thank you again for your very useful and constructive review. Based on your comments, we believe that our manuscript has substantially improved. We are looking forward to receiving your feedback on this newly improved version.

 

7. PLOS authors have the option to publish the peer review history of their article (what does this mean?). If published, this will include your full peer review and any attached files.

Do you want your identity to be public for this peer review? For information about this choice, including consent withdrawal, please see our Privacy Policy.

Reviewer #3: No

Reviewer #4: No

Reviewer #5: No

Reviewer #6: No

Reviewer #7: No

---

## [Decision Letter · Decision Letter 3]

25 Sep 2023

Trajectories of self-kindness, common humanity, and mindfulness during the COVID-19 pandemic: A person-oriented multi-trajectory approach

PONE-D-22-01485R3

Dear Dr. Wollast,

We’re pleased to inform you that your manuscript has been judged scientifically suitable for publication and will be formally accepted for publication once it meets all outstanding technical requirements.

Kind regards,

Giuseppe Marano

Academic Editor

PLOS ONE

Additional Editor Comments (optional):

Reviewers' comments:

Reviewer's Responses to Questions

**Comments to the Author**

1. If the authors have adequately addressed your comments raised in a previous round of review and you feel that this manuscript is now acceptable for publication, you may indicate that here to bypass the “Comments to the Author” section, enter your conflict of interest statement in the “Confidential to Editor” section, and submit your "Accept" recommendation.

Reviewer #3: All comments have been addressed

Reviewer #5: All comments have been addressed

2. Is the manuscript technically sound, and do the data support the conclusions?

Reviewer #3: Yes

Reviewer #5: Yes

3. Has the statistical analysis been performed appropriately and rigorously? 

Reviewer #3: Yes

Reviewer #5: I Don't Know

4. Have the authors made all data underlying the findings in their manuscript fully available?

Reviewer #3: Yes

Reviewer #5: Yes

5. Is the manuscript presented in an intelligible fashion and written in standard English?

Reviewer #3: Yes

Reviewer #5: Yes

6. Review Comments to the Author

Reviewer #3: (No Response)

Reviewer #5: The authors have diligently addressed all the suggested revisions from previous reviewers. The paper contributes a valuable addition to the existing body of literature. I recommend accepting this paper without the need for any additional revisions.

7. PLOS authors have the option to publish the peer review history of their article (what does this mean?). If published, this will include your full peer review and any attached files.

Reviewer #3: No

Reviewer #5: No

---

## [Editor Report · Acceptance letter]

8 Dec 2023

PONE-D-22-01485R3 

Trajectories of self-kindness, common humanity, and mindfulness during the COVID-19 pandemic: A person-oriented multi-trajectory approach 

Dear Dr. Wollast:

I'm pleased to inform you that your manuscript has been deemed suitable for publication in PLOS ONE. Congratulations! Your manuscript is now with our production department. 

Kind regards, 

on behalf of

Dr. Giuseppe Marano 

Academic Editor

PLOS ONE